# Tumor necrosis factor mediates USE1-independent FAT10ylation under inflammatory conditions

Leonie Schnell[1,2,*], Alina Zubrod[1,2], Nicola Catone[1,2], Johanna Bialas[1,2], Annette Aichem[1,2,*]

The ubiquitin-like modifier FAT10 is up-regulated in many different cell types by IFNγ and TNFα (TNF) and directly targets proteins for proteasomal degradation. FAT10 gets covalently conjugated to its conjugation substrates by the E1 activating enzyme UBA6, the E2 conjugating enzyme USE1, and E3 ligases including Parkin. To date, USE1 was supposed to be the only E2 enzyme for FAT10ylation, and we show here that a knockout of USE1 strongly diminished FAT10 conjugation. Remarkably, under inflammatory conditions in the presence of TNF, FAT10 conjugation appears to be independent of USE1. We report on the identification of additional E2 conjugating enzymes, which were previously not associated with FAT10. We confirm their capacity to be charged with FAT10 onto their active site cysteine, and to rescue FAT10 conjugation in the absence of USE1. This finding strongly widens the field of FAT10 research by pointing to multiple, so far unknown pathways for the conjugation of FAT10, disclosing novel possibilities for pharmacological interventions to regulate FAT10 conjugation under inflammatory conditions and/or viral infections.

## Introduction

The posttranslational modification of proteins with ubiquitin or ubiquitin-like modifiers (ULMs) is a mechanism that regulates various cellular pathways such as the stability, function, or subcellular localization of proteins (Kerscher et al, 2006). The covalent attachment of ubiquitin to a substrate protein is mediated by an enzymatic cascade involving an E1 activating enzyme, an E2 conjugating enzyme, and an E3 ligase. E3 ligases mediate the final step of the conjugation, resulting in the formation of an isopeptide linkage between the C-terminal glycine of ubiquitin and the ε-amino group of an internal lysine residue of the conjugation substrate (Hershko & Ciechanover, 1998; Kerscher et al, 2006; Finley, 2009). To date, two E1 activating enzymes for ubiquitin are described, namely, UBA6 (also called UBE1L2, E1-L2, or MOP-4) (Chiu

et al, 2007; Jin et al, 2007; Pelzer et al, 2007) and UBE1 (Ciechanover et al, 1981), dozens of E2 conjugating enzymes (Jin et al, 2007), and hundreds of E3 ligases. E2 conjugating enzymes share a highly conserved catalytic ubiquitin-conjugating (UBC) domain; however, they differ in their specific N- or C-terminal extensions and are therefore classified into four large groups (van Wijk et al, 2009). They are further discriminated as constitutively active or as regulated E2 enzymes, depending on the presence of a conserved phosphorylation site nearby the active site cysteine (Valimberti et al, 2015).

Besides ubiquitin, several so-called ULMs are described, such as small ULM (SUMO1/2/3) (Flotho & Melchior, 2013; Pichler et al, 2017), interferon-stimulated gene 15 (ISG15) (Albert et al, 2018; Dzimianski et al, 2019), or HLA-F adjacent transcript 10 (FAT10; also known as UBD) (Fan et al, 1996). FAT10 consists of two ubiquitin-like domains, which are arranged in a tandem array and connected by a short flexible linker of five amino acids (Fan et al, 1996; Theng et al, 2014; Aichem et al, 2018). While ubiquitin is ubiquitously expressed, a constitutive FAT10 expression is restricted to cells of the immune system such as mature dendritic cells, CD8[+] T cells, natural killer cells, natural killer T cells, or medullary thymic epithelial cells (Bates et al, 1997; Liu et al, 1999; Lukasiak et al, 2008; Buerger et al, 2015; Schregle et al, 2018). Importantly, FAT10 mRNA and protein expressions are highly up-regulated in almost all cell types upon exposure to the proinflammatory cytokines IFNγ and TNFα, pointing to a specific role of FAT10 in immune regulation and inflammation (Liu et al, 1999; Raasi et al, 1999; Aichem et al, 2010; Choi et al, 2014; Buerger et al, 2015; Mah et al, 2019). Likewise, FAT10 is highly up-regulated in several cancer types, most probably mediated by the inflammatory tumor microenvironment (Lukasiak et al, 2008; Ji et al, 2009; Gao et al, 2014; Aichem & Groettrup, 2016). The earlier hypothesis that each ULM uses its private set of E1, E2, and E3 enzymes was disproved with the discovery of the FAT10 E1 activating enzyme UBA6, the FAT10 E2 conjugating enzyme UBA6-specific E2 enzyme 1 (USE1), and the recently discovered FAT10 E3 ligase Parkin, because these enzymes are bispecific and activate and transfer not only FAT10 but also ubiquitin (Chiu et al, 2007; Jin et al, 2007; Pelzer et al, 2007; Aichem et al, 2010; Roverato et al, 2021). Although many E2 conjugating enzymes can be loaded with either UBA6- or UBE1-

[1]Biotechnology Institute Thurgau at the University of Konstanz, Kreuzlingen, Switzerland   [2]Division of Immunology, Department of Biology, University of Konstanz, Konstanz, Germany

Correspondence: Annette.Aichem@bitg.ch
*Leonie Schnell and Annette Aichem contributed equally to this work

activated ubiquitin onto their active site cysteine (Jin et al, 2007), USE1 was shown to be exclusively loaded with UBA6-activated ubiquitin or FAT10 (Jin et al, 2007; Aichem et al, 2010). This finding together with data showing that siRNA-mediated knockdown of USE1 strongly diminished bulk FAT10 conjugation let us suggest for a long time that USE1 might be the main if not the only E2 conjugating enzyme for FAT10 conjugation (Aichem et al, 2010). In the present study, we provide strong evidence for changing this paradigm by identifying several E2 conjugating enzymes, which can replace USE1 as an E2 conjugating enzyme for FAT10 conjugation. Although a CRISPR/Cas9-based knockout (ko) of USE1 almost completely abolished FAT10 conjugation under non-inflammatory conditions, the addition of TNF or the overexpression of most of the newly identified E2 conjugating enzymes was sufficient to rescue FAT10 conjugation in the absence of USE1. This provides strong evidence that additional E2 conjugating enzymes can mediate FAT10 conjugation either under non-inflammatory or under inflammatory conditions in the presence of TNF. Using a transcriptome analysis in combination with biochemical screening of known UBA6-interacting E2 conjugating enzymes, seven E2 conjugating enzymes were identified, which all can accept UBA6-activated FAT10 in the absence of USE1.

Collectively, we show that the E2 usage for FAT10ylation is not restricted to USE1 and that different E2 conjugating enzymes contribute to FAT10 conjugation in a cell type– and TNF-dependent manner. Our results imply that mechanistically and functionally, FAT10ylation in steady-state immune cells differs from that under inflammatory conditions by differences in the E2 usage. In support of this hypothesis, we present two new FAT10 conjugation substrates, which are FAT10ylated either in a USE1-independent manner or in dependence of TNF in USE1 knockout cells. Moreover, our results point to a variety of so far unknown FAT10 conjugation pathways and might help to understand how FAT10ylation is regulated when constitutively expressed in immune cells or when induced by proinflammatory cytokines under inflammatory conditions.

# Results

### Knocking out USE1 points to a minor role of USE1 in FAT10 conjugation under inflammatory conditions

The E2 conjugating enzyme USE1 was published already several years ago by our group as an E2 conjugating enzyme for FAT10 conjugation (Aichem et al, 2010). Because siRNA-mediated knockdown of USE1 never completely abolished USE1 expression on the protein level (Aichem et al, 2010), the CRISPR/Cas9 system was used to create a USE1 knockout (ko) cell line in human HEK293 cells (Aichem et al, 2018). To confirm the importance of USE1 as the E2 conjugating enzyme in the FAT10ylation cascade, HEK293-USE1-ko cells were transiently transfected with the mammalian expression construct pcDNA3.1-His-3xFLAG-FAT10 (Chiu et al, 2007) to express a His-3xFLAG-tagged version of FAT10 (from now on referred to as FLAG-FAT10). As a control, two independent knockout cell clones of the FAT10 E1 activating enzyme UBA6 were included. To monitor

FLAG-FAT10 conjugate formation, cells were directly lysed in denaturing gel sample buffer and subjected to Western blot analysis using FLAG-reactive antibodies. As shown in Fig 1A, FLAG-FAT10–expressing HEK293 WT cells (293) showed the characteristic band pattern of bulk FAT10 conjugates (Fig 1A, lane 2), whereas almost no FLAG-FAT10 conjugates were visible in UBA6-ko or USE1-ko cells (Fig 1A, lanes 3–5), pointing to an irreplaceable role of the two enzymes for FAT10 conjugation under steady-state conditions. To further confirm this finding, the same experiment was repeated under endogenous conditions upon induction of FAT10 expression with the proinflammatory cytokines IFNγ and TNF (Fig 1B). To visualize endogenous FAT10 conjugates, FAT10 was immunoprecipitated using a FAT10-reactive antibody (clone 4F1; Enzo Life Sciences [Aichem et al, 2010]) in combination with a polyclonal FAT10-reactive antibody for the detection by Western blot analysis (Hipp et al, 2004). Although FAT10 conjugates were clearly visible in HEK293 WT cells (Fig 1B, lane 2), FAT10 expression and conjugation were absent in HEK293-FAT10-ko cells and strongly diminished in UBA6-ko cells (Fig 1B, lanes 3–5). In contrast to the results obtained under overexpressing conditions (Fig 1A), treatment of USE1-ko cell lines with IFNγ/TNF rescued FAT10 conjugation almost to the same degree as it was observed in WT cells (Fig 1B, lanes 6, 7 versus lane 2). This let us hypothesize that in HEK293 cells, at least one additional E2 conjugating enzyme for FAT10 conjugation must exist, which might become expressed and/or activated under inflammatory conditions. To investigate whether activation of the putative E2 conjugating enzyme(s) is dependent on both proinflammatory cytokines, or whether either IFNγ or TNF alone might be sufficient to stimulate the expression or activation of the E2 conjugating enzyme(s), HEK293-USE1-ko cells expressing FLAG-FAT10 were left untreated, or in addition stimulated with IFNγ or TNF, either alone or in combination (Fig 1C). Although IFNγ treatment did not rescue FLAG-FAT10 conjugation (Fig 1C, lane 4), TNF alone was sufficient to restore FLAG-FAT10 conjugation to the same extent as when IFNγ and TNF were applied together (Fig 1C, lanes 5, 6). These findings strongly suggested the existence of at least one additional E2 conjugating enzyme whose mRNA expression is induced upon TNF treatment in HEK293 cells, or which gets activated for instance by TNF-mediated phosphorylation (Valimberti et al, 2015). FAT10 expression is highly up-regulated in different types of malignancies such as colon or hepatocellular cancer (Lukasiak et al, 2008). Therefore, we first investigated whether the colon cancer cell line HCT116 and the hepatocellular cancer cell line Huh7 were able to express and to conjugate both endogenous FAT10 upon treatment with IFNγ and TNF (Fig S1A) and overexpressed FLAG-tagged FAT10 (Fig S1B). Thereafter, both cell lines were used to knock out USE1 expression using the CRISPR/Cas9 technology. Confirming our results obtained from HEK293 cells, USE1 deficiency strongly diminished FAT10 conjugation in Huh7-USE1-ko cells as compared to Huh7 WT cells (Fig 1D, lane 5 versus 2) and FAT10 conjugation was partially rescued in the presence of TNF or upon the overexpression of 6His-USE1 (His-USE1) (Fig 1D, lanes 6, 7). In contrast, knocking out USE1 in two independent HCT116-USE1-ko clones had only a minor effect on bulk FAT10 conjugation and treatment with TNF did not affect FAT10 conjugation, suggesting that USE1 plays only a minor or redundant role in this cell line (Fig 1E).

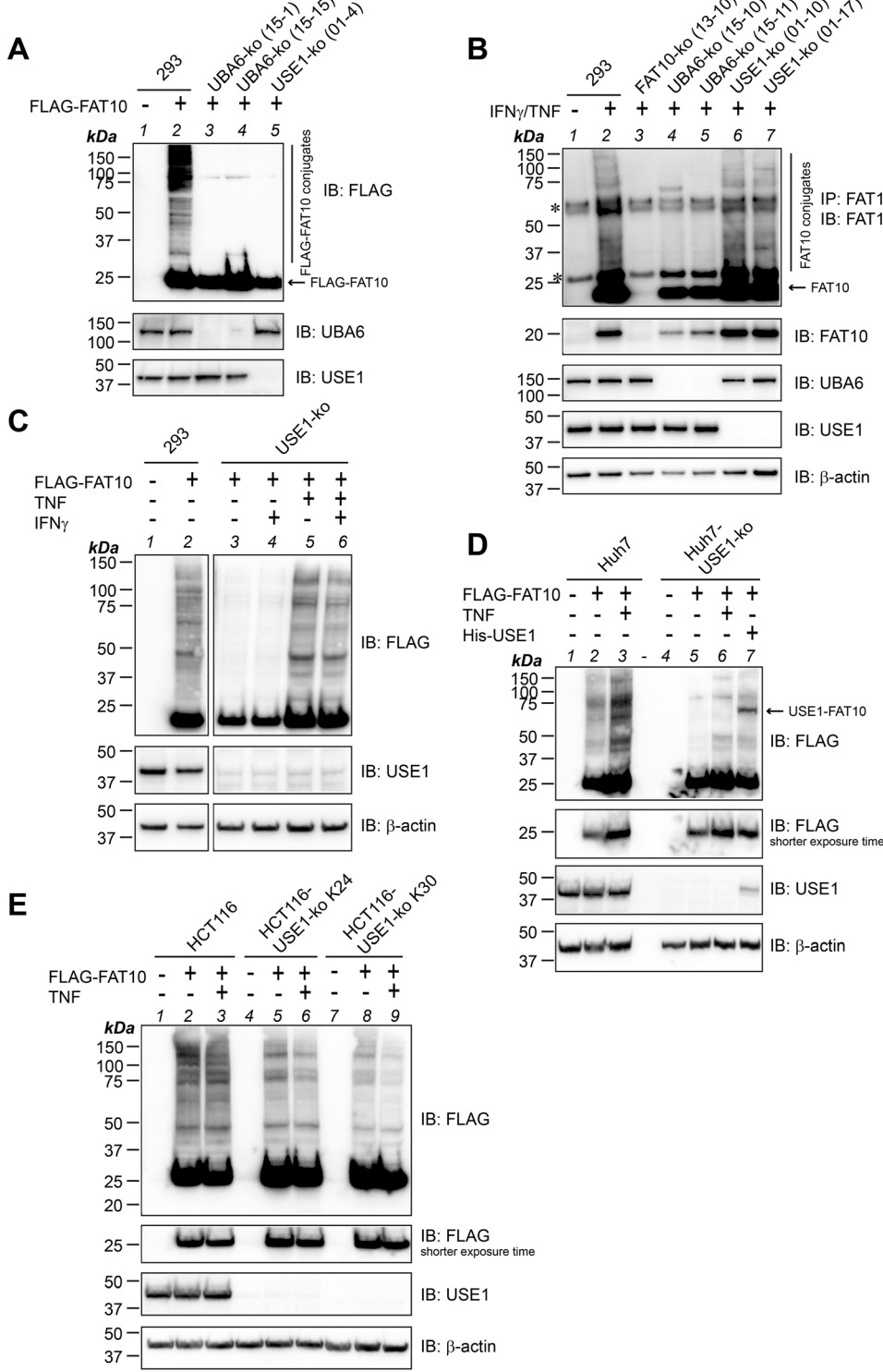

**Figure 1. TNF treatment restores FAT10 conjugation in USE1-ko cells.**
**(A)** FLAG-FAT10 conjugation was monitored in HEK293 WT (293), CRISPR/Cas9-based UBA6, or USE1 knockout (ko) cells (UBA6-ko clones 15-1 and 15-15, USE1-ko cell clone 01-4). Cells were transiently transfected with an expression plasmid for His-3xFLAG-FAT10 (FLAG-FAT10), and crude lysates were prepared under denaturing conditions. SDS–PAGE and Western blot analysis (IB) were performed with FLAG-reactive antibodies to visualize bulk FAT10 conjugates. UBA6- and USE1-reactive polyclonal antibodies were used to confirm the knockout of the two proteins. **(B)** HEK293 WT (293), FAT10-ko (clone 13-10), UBA6-ko (clones 15-10 and 15-11), and USE1-ko (clones 01-10 and 01-17) cells were treated for 24 h with IFNγ and TNF to induce endogenous FAT10 expression. Cells were lysed under denaturing conditions and subjected to immunoprecipitation using a FAT10-reactive monoclonal antibody (clone 4F1). Endogenous FAT10 conjugates were visualized with a polyclonal FAT10-reactive antibody. Specific antibodies reactive to UBA6 or USE1 were used for the confirmation of the respective knockout. β-Actin was used as a loading control. Asterisks mark the heavy and light chain of the antibody used for immunoprecipitation. **(C)** HEK293 WT (293) or USE1-ko cells were transiently transfected with an expression construct for FLAG-FAT10 and subsequently treated with either TNF or IFNγ, both together, or left untreated for 24 h. Crude lysates were prepared, and proteins were analyzed with the antibodies indicated. β-Actin was used as a loading control. **(D)** Huh7 WT or USE1-ko cells were transiently transfected with expression constructs for FLAG-FAT10 and/or His-USE1, as indicated, and subsequently treated for 24 h with TNF. Crude lysates were prepared and analyzed as described in (A). **(E)** HCT116 WT or USE1-ko cells (clone K24 or K30) transiently expressing FLAG-FAT10 were treated for 24 h with TNF, as indicated. Crude cell lysates were prepared, and FLAG-FAT10 conjugation was analyzed as described in (A). All experiments were performed at least three times with similar outcomes.
Source data are available for this figure.

In summary, upon IFNγ/TNF treatment, USE1 appeared to be dispensable for FAT10 conjugation in certain cell lines. TNF alone was sufficient to rescue FAT10 conjugation in HEK293 and Huh7 cells, indicating that additional E2 conjugating enzymes for FAT10 conjugation must exist, which are either induced on the transcriptional level, stabilized, or activated by posttranslational modifications, for example, by TNF-dependent phosphorylation, as described for several E2 conjugating enzymes (Valimberti et al, 2015). Because a knockout of USE1 in HCT116 cells did not profoundly affect FAT10 conjugation, we suggested that FAT10

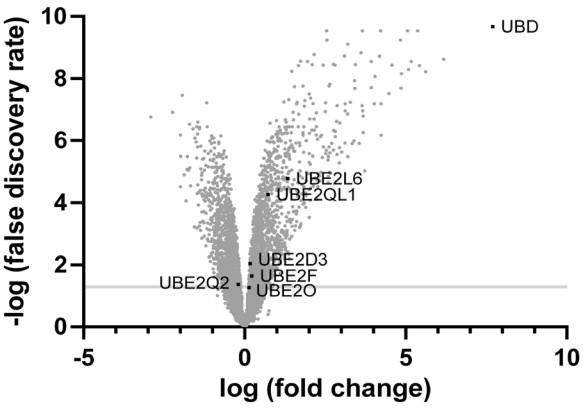

**Figure 2. Transcriptome analysis of TNF-treated HEK293 cells identifies significantly regulated E2 conjugating enzymes.**
HEK293 WT cells were treated for 24 h with TNF or left untreated. mRNA was isolated and subjected to a transcriptome analysis as described in the Materials and Methods section, using three independent samples per condition and a false discovery rate of < 5%. As a positive control for the successful treatment with TNF, the mRNA expression level of FAT10 (UBD) was labeled.

conjugation might be mediated by additional, previously uncharacterized E2 conjugating enzymes also under non-inflammatory conditions, for example, in certain immune cells.

### A transcriptome analysis identifies five TNF-regulated E2 conjugating enzymes

To identify TNF-regulated E2 conjugating enzymes, we performed a transcriptome analysis to discover genes, which were significantly up-regulated on their mRNA level in HEK293 cells, which were treated for 24 h with TNF in comparison with untreated cells (Fig 2). By applying a false discovery rate of < 5%, five E2 conjugating enzymes were identified, which were meeting these criteria. Four of the E2 enzymes were significantly up-regulated, namely, UBE2L6 (also called UBCH8), UBE2QL1, UBE2D3 (also called UBCH5C), and UBE2F (Fig 2 and Table 1). However, as compared to FAT10 (UBD), which was up-regulated by about 209-fold upon treatment with

TNF, the mRNAs of the four E2 conjugating enzymes were increased only to a minor extent. For example, although the ISG15-specific E2 conjugating enzyme UBE2L6 was up-regulated by about 2.5-fold, all other E2 mRNA levels were up-regulated only by about 1.1–1.7-fold (Fig 2 and Table 1). Thus, we decided to include also the E2 conjugating enzyme UBE2O with a false discovery rate of 0.054 (1.098-fold regulated), as well as UBE2Q2 in further analyses. Of note, UBE2Q2 was even slightly down-regulated upon TNF treatment (0.85-fold) and thus considered as a negative control (Table 1).

### UBE2Q2 and UBE2QL1 are new FAT10 conjugation substrates

After the identification of TNF-regulated E2 conjugating enzymes by our transcriptome analysis, we went further to prove whether these E2 enzymes might be able to act as FAT10-specific E2 enzymes. Hence, we investigated under in vitro and/or in cellulo conditions whether FAT10, activated by UBA6, could be loaded onto the active site cysteine of these E2 conjugating enzymes. Because we did not manage to express decent amounts of UBE2F, the known E2 conjugating enzyme for the ULM NEDD8 (Huang et al, 2009), we excluded this E2 enzyme from further experiments. UBE2L6 was not loaded with FAT10, neither upon overexpression in HEK293-USE1-ko cells, nor under in vitro conditions with recombinant proteins (Fig S2A and B), whereas as a positive control, a transfer of recombinant ISG15 onto UBE2L6 was detectable (Fig S2B, lane 6). In order not to miss putative activation of UBE2L6 by TNF, we in addition treated the cells for 24 h with TNF before harvesting. However, still no FAT10 loading onto UBE2L6 was observed (Fig S2A, lane 8), confirming the high specificity of UBE2L6 for ISG15 conjugation (Durfee & Huibregtse, 2012). HA-tagged UBE2Q2 and UBE2QL1 were both loaded with FLAG-tagged FAT10 but not with its conjugation-deficient mutant FAT10-AV, in which the C-terminal diglycine motif was exchanged by alanine and valine (-AV) (Fig S2C, IP: HA, IB: FLAG, lanes 6, 7 and 8, 9). Interestingly, these signals were observed only in TNF-treated HEK293-USE1-ko cells as shown in Fig S2C, whereas in untreated cells, these FAT10 conjugates were not visible (data not shown). To further investigate whether FAT10 becomes thioester bound to the respective active site cysteine of either of the two E2 enzymes, site-directed mutagenesis was

**Table 1. Significantly up- or down-regulated mRNA expression levels of E2 conjugating enzymes in HEK293 cells treated with TNF.**

| E2 conjugating enzyme | Alternative name | Loaded with UBA6-activated ubiquitin | False discovery rate < 5% | Fold up-regulated on the mRNA level ($2^{\log2\ ratio}$) |
|---|---|---|---|---|
| UBE2L6 | ubiquitin/ISG15-conjugating enzyme E2 L6; UBCH8 | no, Jin et al (2007) | $1.6 \times 10^{-5}$ | 2.5 |
| UBE2QL1 | ubiquitin-conjugating enzyme E2 Q family like 1 | n.d. | $5.5 \times 10^{-5}$ | 1.65 |
| UBE2D3 | ubiquitin-conjugating enzyme E2 D3; UBCH5C | yes, Jin et al (2007) | 0.0089 | 1.12 |
| UBE2F | ubiquitin-conjugating enzyme E2 F (putative); NEDD8-conjugating enzyme | n.d. | 0.02 | 1.16 |
| UBE2O | ubiquitin-conjugating enzyme E2-230K | yes, this work | 0.054 | 1.098 |
| UBE2Q2 | ubiquitin-conjugating enzyme E2 Q2 | no, Jin et al (2007) | 0.04 | 0.87 |

Total mRNA was isolated from HEK293 WT cells, treated or not for 24 h with 600 U/ml TNF. mRNA expression levels of E2 conjugating enzymes were analyzed for their significant up- or down-regulation, as compared to the untreated control. A false discovery rate of < 5% was applied. As a positive control, FAT10 (UBD) mRNA expression was measured (false discovery rate: $2.12 \times 10^{-10}$, corresponding to a 209-fold up-regulation). n.d., not determined.

performed to create the catalytically dead mutants HA-UBE2Q2-C304A and HA-UBE2QL1-C88A, respectively. However, FAT10 was still conjugated to both active site cysteine mutants under non-reducing conditions (without *β*-2-ME) (Fig S2D, IP: HA, IB: FLAG, lanes 5, 6 and 9, 10) and under reducing conditions in the presence of 4% 2-ME (data not shown), pointing to FAT10 conjugation substrates rather than to a function as FAT10 E2 conjugating enzymes.

**UBE2D3 and UBE2O are new FAT10 E2 conjugating enzymes**

In the next step, the two remaining E2 conjugating enzymes identified in the transcriptome analysis were investigated, namely, UBE2D3 and UBE2O. In contrast to the results obtained above, a transfer of FAT10 onto the E2 conjugating enzymes UBE2D3 and UBE2O could be confirmed both under in cellulo and under in vitro conditions. An in vitro experiment with recombinant FLAG-UBA6, 6His-UBE2D3 (His-UBE2D3), its active site cysteine mutant His-UBE2D3-C85A, and HA-tagged FAT10 C0 (C134L) was performed. The stabilized version of FAT10, HA-FAT10 C0 (C134L) (Aichem et al, 2018), was used to diminish background signals, which often appear when using recombinant untagged FAT10. Under non-reducing conditions, recombinant His-UBE2D3 was loaded in an ATP-dependent manner with FAT10 (Fig 3A, IB: HA, IB: His [non-red.], lanes 6, 7 and 17, 18), and the UBE2D3-FAT10 conjugate was absent when the WT His-UBE2D3 was exchanged for the active site cysteine mutant His-UBE2D3-C85A (Fig 3A, IB: HA, IB: His [non-red.], lanes 10 and 21). The UBE2D3-FAT10 conjugate almost completely disappeared under reducing conditions in the presence of 4% 2-ME (Fig 3A, IB: HA, IB: His [red.], lanes 6, 7 and 17, 18), pointing to a thioester bond formation between FAT10 and the active site cysteine of UBE2D3. This finding was further confirmed under in cellulo conditions in HEK293-USE1-ko cells, upon the overexpression of HA-tagged UBE2D3 or its active site cysteine mutant in combination with FLAG-tagged WT FAT10 or its conjugation inactive mutant FLAG-FAT10-AV (Fig 3B, IP: HA, IB: FLAG, lanes 6–9). Taken together, our results imply that UBE2D3 acts as a FAT10 E2 conjugating enzyme. To further investigate the importance of UBE2D3 in FAT10ylation, HEK293-UBE2D3 knockout and HEK293-USE1/UBE2D3 double-knockout cell lines were generated. A UBE2D3 knockout alone did not diminish bulk FAT10 conjugation as compared to WT HEK293 cells (Fig 3C, IB: FLAG, lanes 2 versus 4), most probably because USE1 was still expressed. In support of this hypothesis, deletion of USE1 and UBE2D3 at the same time clearly reduced FAT10 conjugation to the same extent as observed in USE1 single knockouts (Fig 3C, IB: FLAG, lanes 3 versus 5). This suggests that UBE2D3 might be a FAT10 E2 conjugating enzyme for only a small subset of conjugation substrates, or it might point to a cell type–specific function of UBE2D3 as a FAT10 E2 conjugating enzyme. Finally, the 143-kD E2 conjugating enzyme UBE2O was investigated, bearing N- and C-terminal extensions and belonging to the same group of E2 conjugating enzymes as USE1 (Ullah et al, 2019). UBE2O is described to be an E2/E3 hybrid protein with an active site cysteine for E2 activity (Cys1040) and an additional putative active site cysteine for E3 ligase activity (Cys617) (Yanagitani et al, 2017; Chen et al, 2018). Site-directed mutagenesis was performed to create active site cysteine mutants of the single cysteines (C617A or C1040A), or of both at the same time (C617/1040A). Because purification of recombinant full-length UBE2O did not result in adequate amounts to perform in vitro loading assays, we performed a semi–in vitro

approach and purified truncated FLAG-tagged versions of UBE2O (FLAG-UBE2O trunc and FLAG-UBE2O trunc-C1040A) by FLAG immunoprecipitation from HEK293-UBA6/USE1/UBE2O triple-ko cells. As shown in the cartoon in Fig 3D, UBE2O trunc (amino acids 812–1,292) comprises the coiled-coil domain (CC), the catalytic core domain (UBC) with the active site cysteine C1040, and the remaining C-terminus of UBE2O. FLAG-UBE2O trunc or FLAG-UBE2O trunc-C1040A bound to the antibody beads was incubated with the respective recombinant proteins. Under these semi–in vitro conditions, loading of UBA6-activated FAT10 was observed onto FLAG-UBE2O trunc but not onto catalytically inactive FLAG-UBE2O trunc-C1040A (Fig 3D, IB: HA, lanes 4 versus 8). Moreover, the conjugation-incompetent FAT10 mutant HA-FAT10 C0 (C134L)-GC was binding to none of the UBE2O variants (Fig 3D, IB: HA, lanes 5 and 9). Recombinant 6His-ubiquitin was included as a positive control and was transferred only onto FLAG-UBE2O trunc (Fig 3D, IB: His, lanes 6, 10). To our knowledge, this is the first time where it is shown that UBE2O can interact with UBA6 and accept activated ubiquitin or FAT10 onto its active site cysteine. To confirm a FAT10 transfer onto UBE2O under in cellulo conditions, FLAG-tagged UBE2O WT and active site cysteine mutants were expressed together with HA-FAT10 or HA-FAT10-AV in HEK293-USE1/UBE2O double-knockout cells and FAT10 loading was compared under non-reducing (without 2-ME) and reducing (4% 2-ME) conditions (Fig 3E). Upon immunoprecipitation using FLAG-reactive antibodies, a transfer of HA-FAT10 onto FLAG-UBE2O was observed under non-reducing conditions, and to a lesser extent under reducing conditions (Fig 3E, upper panels, IP: FLAG, IB: HA [non-red. and red.], lane 8). Confirming the results obtained under in vitro conditions, no UBE2O-FAT10 conjugate was visible when HA-FAT10-AV was expressed and a diminished amount of conjugate was formed with the UBE2O-C1040A mutant (Fig 3E, upper panels, IP: FLAG, IB: HA, lanes 9, 10). A mutation of the active site cysteine for the UBE2O E3 ligase activity to alanine, C617A, did not abrogate FAT10 transfer onto UBE2O, whereas the double-mutant UBE2O-C617/1040A again showed slightly diminished FAT10 loading as compared to the WT (Fig 3E, upper panels, IP: FLAG, IB: HA, lanes 11, 12). This suggests that Cys617 might not have FAT10 reactivity, but that Cys1040 does form a reduction-sensitive thioester with FAT10, pointing to the E2 activity of UBE2O for FAT10 conjugation. As observed for UBE2D3, a CRISPR/Cas9-based knockout of UBE2O in HEK293 cells only slightly diminished FLAG-FAT10 conjugation, which again was strongly diminished when both USE1 and UBE2O were knocked out at the same time (Fig 3C, IB: FLAG, lanes 6 and 7 as compared to lanes 2 and 3). In summary, two of the E2 conjugating enzymes, which were identified as TNF-regulated E2 enzymes on the mRNA transcription level, namely, UBE2D3 and UBE2O, were confirmed as FAT10 E2 conjugating enzymes. Because a specific knockout did not abrogate bulk FAT10 conjugation, we suggest that both E2 enzymes might be necessary for some specific FAT10 conjugation substrates or might become active only under inflammatory conditions or in specific cell types or tissues.

**Several additional UBA6-interacting E2 conjugating enzymes perform FAT10ylation in the presence or absence of USE1**

The E2 conjugating enzymes identified in our transcriptome analysis were only slightly, albeit significantly, up-regulated upon

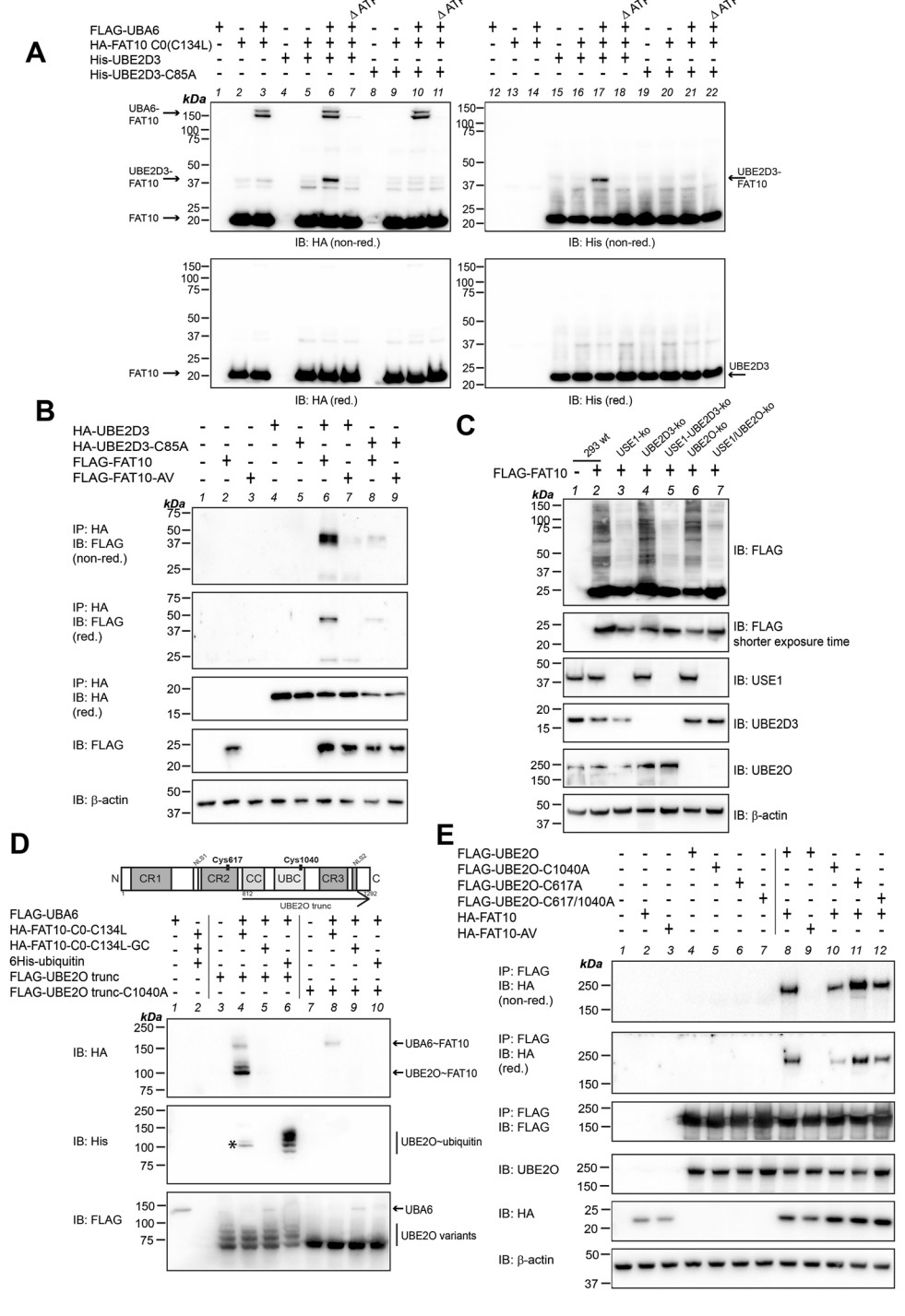

**Figure 3. UBE2D3 and UBE2O accept FAT10 onto their active site cysteine.**

**(A)** In vitro FAT10 loading experiment. Recombinant 6His-tagged UBE2D3 (His-UBE2D3) or its active site cysteine mutant His-UBE2D3-C85A was incubated with FLAG-UBA6 and HA-FAT10 C0 (C134L) in in vitro buffer in the presence or absence of ATP, as indicated. Proteins were incubated for 30 min at 37°C, and reactions were stopped by the addition of 5x gel sample buffer and subsequent boiling. Reactions were applied to SDS–PAGE and Western blot analysis using HA- or His-reactive antibodies, under non-reducing (non-red.) or reducing (4% 2-ME, red.) conditions. **(B)** HEK293-USE1-ko cells were transiently transfected with expression constructs for HA-UBE2D3, HA-UBE2D3-C85A, FLAG-FAT10, or its conjugation-incompetent mutant FLAG-FAT10-AV, as indicated. After 24 h, cells were lysed and cleared lysates were subjected to immunoprecipitation (IP) using HA-reactive antibodies. UBE2D3-FAT10 conjugates were analyzed under non-reducing (non-red.) or reducing (4% 2-ME, red.) conditions, using the antibodies indicated. β-Actin was used as a loading control. **(C)** HEK293 WT, USE1-ko, UBE2D3-ko, USE1/UBE2D3-ko, UBE2O-ko, or USE1/UBE2O-ko cells were transiently transfected with an expression construct for FLAG-FAT10. Crude lysates were prepared under denaturing conditions, and proteins were subjected to SDS–PAGE and Western blot analysis, using the antibodies indicated. β-Actin was used as a loading control. All experiments were performed at least three times with similar outcomes. **(D)** Cartoon shows the domain structure of UBE2O with the conserved regions 1–3 (CR1-3 domains), coiled-coil domain (CC), the ubiquitin core catalytic domain (UBC), and two putative nuclear localization signals, as described in Hormaechea-Agulla et al (2018). The Western blot shows a semi–in vitro FAT10 loading experiment. Truncated FLAG-tagged UBE2O WT (FLAG-UBE2O trunc), as indicated in the cartoon, or its active site cysteine mutant FLAG-UBE2O trunc-C1040A cells were purified from transiently transfected HEK293-UBA6/USE1/UBE2O triple-knockout cells by immunoprecipitation, using FLAG-reactive antibodies. Beads were washed intensively, and proteins were left bound to the beads. Recombinant proteins were added as indicated, and reactions were incubated for 30 min at 37°C. Reactions were stopped by the addition of 5x gel sample buffer and subsequent boiling. Proteins were subjected to SDS–PAGE and Western blot analysis under non-reducing conditions using

the antibodies indicated. An asterisk marks stripping leftover from the Western blot, shown in the upper panel (IB: HA). **(E)** HEK293-USE1/UBE2O double-knockout cells were transiently transfected with expression constructs for the proteins indicated. 24 h later, cells were lysed and cleared lysates were subjected to immunoprecipitation using FLAG-reactive antibodies. Loading of FAT10 onto UBE2O variants was analyzed under non-reducing (non-red.) or reducing (4% 2-ME, red.) conditions with the antibodies indicated. β-Actin was used as a loading control.
Source data are available for this figure.

TNF treatment. Moreover, FAT10 conjugation in HCT116-USE1-ko cells was only slightly diminished pointing to a rather dispensable role for USE1 in HCT116 cells (Fig 1E). Thus, we suggested that the E2 conjugating enzyme(s) which could rescue FAT10 conjugation in USE1-ko cells might not necessarily be up-regulated by TNF on the mRNA level, but may already be expressed and activated upon TNF treatment.

**Table 2.  Additional E2 conjugating enzymes screened for their ability to be loaded with FAT10.**

| E2 conjugating enzyme | Alternative name | In vitro loading with FAT10 | In cellulo loading with FAT10 | Reconstitution of FAT10 conjugation in USE1-ko cells |
|---|---|---|---|---|
| UBE2A | RAD6A | yes | yes | yes |
| UBE2B | RAD6B | yes | yes | yes |
| UBE2C | UbcH10 | yes | yes | no |
| UBE2D1 | UbcH5A | yes | yes | yes |
| UBE2D2 | UBcH5B | yes | n.d. | not expressed |
| UBE2D3 | UBcH5C | yes | yes | yes |
| UBE2D4 | UBcH5D | no | n.d. | n.d. |
| UBE2E1 | UBCH6 | yes | n.d. | not expressed |
| UBE2E2 | UBCH8 | no | n.d. | n.d. |
| UBE2E3 | UBCH9 | no | n.d. | n.d. |
| UBE2G1 | UBC7 | no | n.d. | n.d. |
| UBE2G2 | UBC7 | yes | yes | yes |
| UBE2T | FANCT | no | n.d. | n.d. |

n.d., not determined; not expressed: no protein expression was detectable by Western blot analysis.

Therefore, we screened additional E2 conjugating enzymes for their ability to accept UBA6-activated FAT10 onto their active site cysteine. As listed in Table 2, twelve additional E2 conjugating enzymes were screened for their ability to accept FAT10 under in vitro and/or in cellulo conditions. The E2 enzymes were chosen based on an earlier publication by Jin and colleagues (Jin et al, 2007) who had screened several E2 conjugating enzymes for their ability to take over ubiquitin not only from UBE1 but also from UBA6 as an E1 activating enzyme. For in vitro experiments, all twelve E2 conjugating enzymes were purified from *E. coli* and applied to in vitro loading experiments with recombinant FLAG-UBA6, untagged FAT10 or FAT10-AV, and/or ubiquitin as a positive control, under non-reducing and/or reducing conditions (Figs 4 and 5). We first focused on UBE2A belonging to class I of E2 conjugating enzymes, which do not contain N- or C-terminal extensions adjacent to the UBC domain (van Wijk et al, 2009). Under in vitro conditions, FAT10 was transferred onto the active site cysteine of UBE2A and the conjugate was almost completely reducible in the presence of 4% 2-ME (Fig 4A, lanes 2 and 8), whereas no FAT10 conjugate was seen when UBE2A-C88A or FAT10-AV was applied instead (Fig 4A, lanes 3, 4 and 9, 10). This finding was further confirmed under non-reducing conditions in HEK293-USE1-ko cells transiently expressing HA-UBE2A or its active site cysteine mutant HA-UBE2A-C88A in combination with FLAG-tagged FAT10 or FAT10-AV (Fig 4B, lanes 6–9). Loading of FAT10 onto UBE2C was observed with both WT and active site cysteine mutants of UBE2C under in vitro conditions (Fig 4C, IB: His, longer exposure time, lanes 7, 8). Therefore, as a control for the correct folding and thus for the functionality of recombinant His-tagged UBE2C, ubiquitin was included as a positive control. As expected, ubiquitin was transferred only onto His-UBE2C but not onto its active site cysteine mutant His-UBE2A-C114A (Fig 4C, lanes 5 and 6). However, because the putative UBE2C-FAT10 signal observed under in vitro conditions was neglectable, it might rather represent an unspecific background signal. This assumption was supported in the following experiments where a transfer of FAT10, but not of FAT10-AV, onto WT

UBE2C but not onto the active site cysteine mutant UBE2C-C114A was confirmed in HEK293-USE1-ko cells (Fig 4D, IP: HA, IB: FLAG, lanes 5–9). Interestingly, the same was observed in case of UBE2G2 where under in vitro conditions, FAT10 was transferred onto both UBE2G2 and its active site mutant HA-UBE2G2-C89A (Fig 4E, IB: His, lanes 7, 8). Similarly, the in vitro ubiquitin control showed loading only onto the WT protein (Fig 4E, IB: His, lanes 5, 6). However, in HEK293-USE1-ko cells, a prominent UBE2G2-FAT10 conjugate was detectable under non-reducing conditions (without 2-ME), and only a weak signal appeared in case of FAT10-AV or the UBE2G2 active site cysteine mutant UBE2G2-C89A (Fig 4F, IP: HA, IB: FLAG, non-red., lanes 6–9). Under reducing conditions (4% 2-ME), a slight UBE2G2-FAT10 signal remained (Fig 4F, IP: HA, IB: FLAG, red., lane 6), which might represent an auto-FAT10ylated UBE2G2, as described before in case of USE1 (Aichem et al, 2010). For the sake of completeness, UBE2B, UBE2G1, and UBE2T were tested under in vitro conditions, and UBE2B was in addition tested also under in cellulo conditions. As shown in Fig 4G and H, FAT10 was transferred onto UBE2B in a FAT10 diglycine–dependent manner under in vitro and in cellulo conditions (Fig 4G, IB: His, longer exposure time, lane 5, and Fig 4H) and in a UBE2B active site cysteine–dependent manner in HEK293 WT cells (Fig S3). However, no FAT10 transfer was observed onto UBEG1 and UBE2T, and ubiquitin was loaded only onto UBE2T but not onto UBE2G1 (Fig 4G, IB: His, lanes 2, 3 and 6, 7). Further screening of all four members of the UBE2D family revealed a FAT10 loading onto UBE2D1 and UBE2D2 and again onto UBE2D3, but not onto UBE2D4 (Fig 5A, IB: His, lanes 3, 5, 7, and 9). In addition to the results obtained for UBE2D3 (Fig 3A and B), FAT10 diglycine– and UBE2D1 active site cysteine–dependent loading of HA-tagged UBE2D1 was likewise confirmed in HEK293-USE1-ko cells (Fig 5B). Last but not least, a very weak UBE2E1-FAT10 conjugate was observable, whereas no loading onto UBE2E2 and UBE2E3 was detectable under in vitro conditions (Fig 5C, IB: His, longer exposure time, lanes 3, 5, and 7). As a control, ubiquitin was transferred onto each of the E2 enzymes of the UBE2E family (Fig 5C, IB: His, lanes 2, 4, and 6). To confirm that FAT10 was

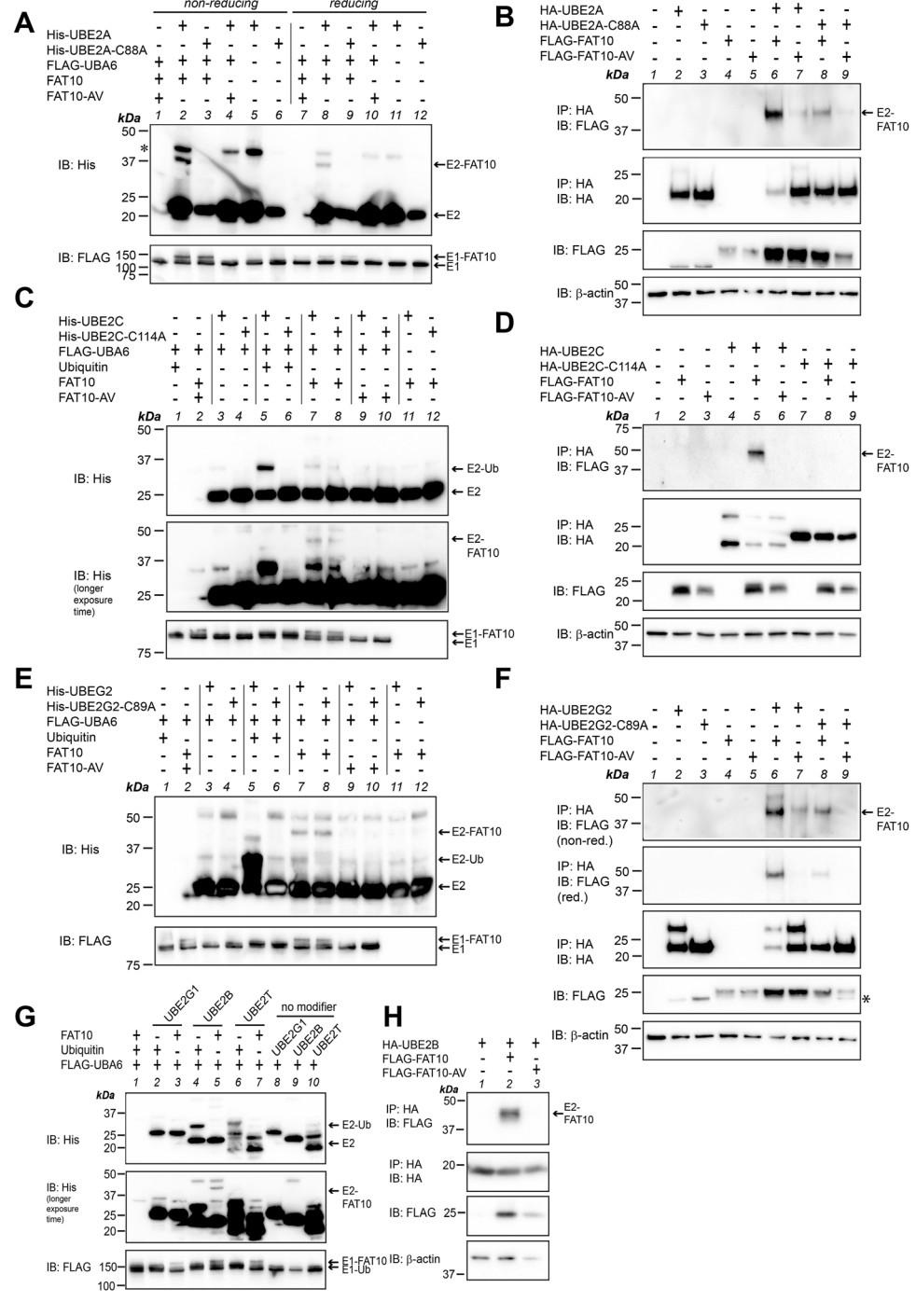

**Figure 4. UBE2A, UBE2C, and UBE2G2 are E2 conjugating enzymes for FAT10ylation.**
**(A)** Recombinant 6His-tagged UBE2A (His-UBE2A), or its active site cysteine mutant His-UBE2A-C88A, was purified from *E. coli* and applied to in vitro loading experiments with recombinant FLAG-UBA6, FAT10, or FAT10-AV, as indicated. Reactions were incubated for 30 min at 37°C and stopped by the addition of 5x gel sample buffer and boiling. Proteins were separated on 12.5% Laemmli gels and subjected to Western blot analysis (IB) under non-reducing (without 2-ME) or reducing (4% 2-ME) conditions using the antibodies indicated. An asterisk marks an unspecific background band. **(B)** HEK293-USE1-ko cells were transiently transfected with expression constructs for HA-UBE2A, HA-UBE2A-C88A, FLAG-FAT10, or FLAG-FAT10-AV, as indicated. After 24 h, cells were harvested and lysed. Cleared cell lysates were subjected to immunoprecipitation (IP) using HA-reactive antibodies. Proteins were visualized by Western blot analysis under non-reducing (without 2-ME) conditions using HA- or FLAG-reactive antibodies. *β*-Actin was used as a loading control. **(C)** Recombinant 6His-tagged UBE2C (His-UBE2C), or its active site cysteine mutant His-UBE2C-C114A, was purified from *E. coli* and applied to in vitro loading experiments. Loading of FAT10 was visualized by Western blot analysis under non-reducing conditions, as described in (A). Recombinant ubiquitin served as a positive control. **(D)** HEK293-USE1-ko cells were transiently transfected with expression constructs for HA-UBE2C, HA-UBE2C-C114A, FLAG-FAT10, or FLAG-FAT10-AV, as indicated. After 24 h, cells were harvested and lysed, and loading of FAT10 was analyzed under non-reducing conditions, as described in (B). **(E)** Recombinant 6His-tagged UBE2G2 (His-UBE2G2), or its active site cysteine mutant 6His–UBE2G2-C89A, was purified from *E. coli* and applied to in vitro loading experiments as described in (A). Recombinant ubiquitin served as a positive control. Loading was monitored by Western blot analysis under non-reducing conditions, as described in (A). **(F)** HEK293-USE1-ko cells were transiently transfected with expression constructs for HA-UBE2G2, HA-UBE2G2-C89A, FLAG-FAT10, or FLAG-FAT10-AV, as indicated. After 24 h, cells were harvested, lysed, and analyzed under non-reducing (non-red.) or reducing (red.) (4% 2-ME) conditions, as described in (B). An asterisk marks an unspecific signal derived from blot stripping. **(G)** Recombinant 6His-tagged UBE2G1, UBE2B, or UBE2T was purified from *E. coli* and applied to in vitro loading experiments using recombinant FAT10 and ubiquitin. Loading onto the respective E2 enzyme was visualized by Western blot analysis under non-reducing conditions, as described in (A). **(H)** HEK293-USE1-ko cells were transiently transfected with expression constructs for HA-UBE2B, FLAG-FAT10, or FLAG-FAT10-AV, as indicated. After 24 h, cells were harvested, lysed, and analyzed under non-reducing conditions, as described in (B). All experiments were performed at least three times with similar outcomes.
Source data are available for this figure.

transferred onto the active site cysteines of these E2 conjugating enzymes not only in the absence but also in the presence of USE1, in cellulo FAT10 loading onto UBE2A, UBE2B, UBE2C, UBE2D1, UBE2D3, UBE2G2, and UBE2O was repeated and confirmed in HEK293 WT cells

(Fig S3). UBE2E1 expression levels were always too low, and therefore, this E2 was not included in the analysis in HEK293 WT cells.

In summary, seven of the twelve additionally screened E2 conjugating enzymes were loaded with FAT10, in dependence of

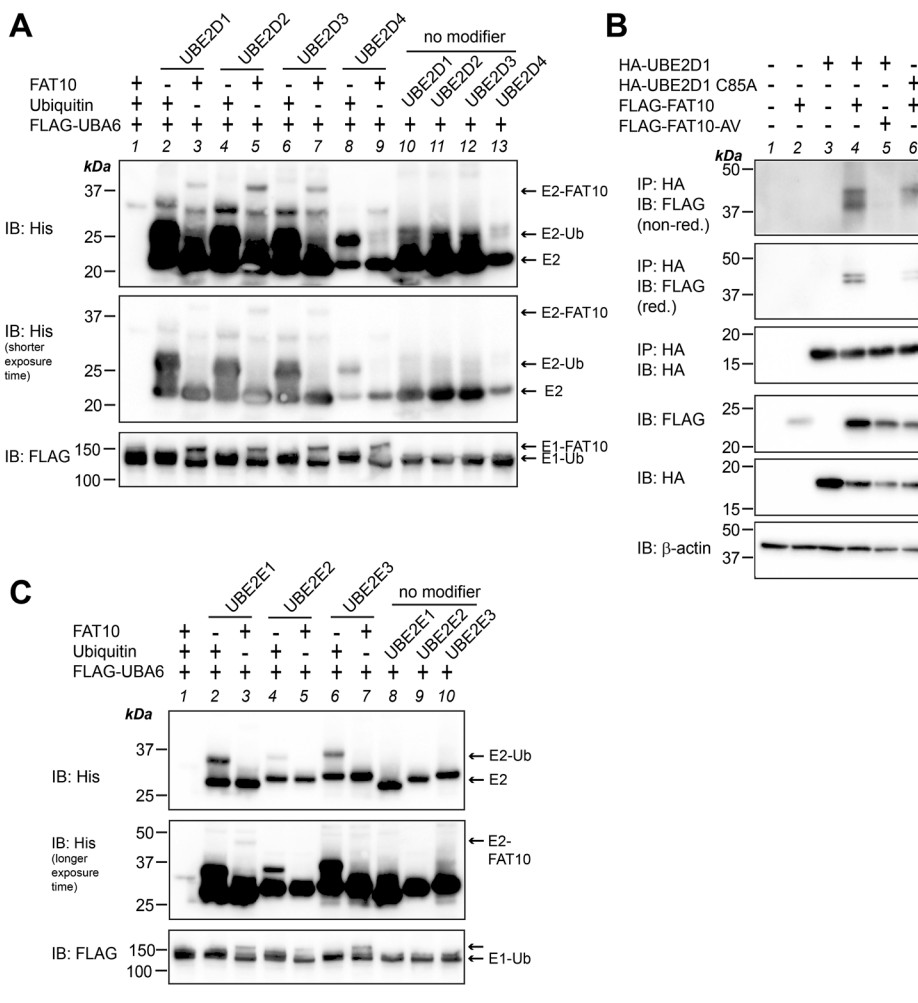

**Figure 5. In vitro and in cellulo experiments confirm loading of FAT10 onto additional E2 conjugating enzymes.** **(A, C)** Recombinant His-tagged E2 conjugating enzymes were incubated with recombinant FAT10 or ubiquitin, as indicated. Reactions were incubated for 30 min at 37°C and stopped by the addition of 5x gel sample buffer and boiling. Reactions were subjected to SDS–PAGE and subsequent Western blot analysis under non-reducing conditions using the antibodies indicated. Shown is one representative experiment each out of three replicates with similar outcomes. **(B)** HEK293-USE1-ko cells were transiently transfected with expression constructs for HA-UBE2D1, HA-UBE2D1 C85A, FLAG-FAT10, or FLAG-FAT10-AV, as indicated. Cleared cell lysates were used for immunoprecipitation using HA-reactive antibodies. Proteins were separated on 12.5% Laemmli gels and visualized with HA- or FLAG-reactive antibodies, as indicated. β-Actin was used as a loading control. Shown is one representative experiment out of three experiments with similar outcomes. Source data are available for this figure.

both the FAT10 diglycine motif and the active site cysteine of the respective E2 conjugating enzyme (Table 2). Together with the two E2 conjugating enzymes UBE2D3 and UBE2O, which had been identified by our transcriptome analysis, all together nine E2 conjugating enzymes could be identified, which were able to accept activated FAT10 from UBA6 in the presence or absence of USE1.

## Overexpression of FAT10 E2 conjugating enzymes partially rescues FAT10 conjugation in HEK293-USE1-ko cells

So far, all FAT10 E2 conjugating enzymes were tested either by in vitro or by in cellulo loading experiments. Therefore, we were wondering whether the overexpression of the individual E2 conjugating enzymes in USE1 knockout cells would be as potent as treatment with TNF to restore FAT10 conjugation in HEK293-USE1-ko cells. For this purpose, HEK293-USE1-ko cells were transiently transfected with expression plasmids for FLAG-FAT10 and the aforementioned E2 conjugating enzymes. Crude cell lysates were prepared under denaturing conditions, and FLAG-FAT10 conjugation was investigated by Western blot analysis (Fig 6A and B). As a positive control, HEK293-USE1-ko cells were stimulated with TNF to reconstitute FLAG-FAT10 conjugation (Fig 6A and B, IB: FLAG, lanes 2,

3). The overexpression of the HA-tagged UBE2D3 restored FLAG-FAT10 conjugation comparable to TNF treatment (Fig 6A, IB: FLAG, lane 4). In contrast, overexpressed HA-UBE2F, HA-UBE2Q2, HA-UBE2QL1, or FLAG-UBE2L6 was not able to rescue FAT10 conjugation (Fig 6A, IB: FLAG, lanes 5–8), confirming our in vitro and in cellulo loading experiments shown in Figs 3 and S2. The expression of HA-tagged UBE2O likewise restored FLAG-FAT10 conjugate formation, validating also this E2 conjugating enzyme as a bona fide FAT10-reactive E2 conjugating enzyme (Fig 6B, IP: FLAG, lane 4). Performing the same experiment with all other identified E2 conjugating enzymes further confirmed FAT10 E2 conjugating enzyme activity for UBE2A, UBE2B, UBE2D1, and UBE2G2 (Fig 6C, IB: FLAG). The expression levels of UBE2D2 and UBE2E1 were always very low. Thus, no conclusion could be drawn if these E2 conjugating enzymes might be able to restore bulk FAT10 conjugation (Fig 6C, IB: HA, lanes 8, 9). However, we do not want to rule out the possibility that these two E2 conjugating enzymes might still be specific for some single FAT10 substrates, because both were loaded with FAT10 at least under in vitro conditions (Fig 5).

Surprisingly, although FAT10 was clearly loaded onto UBE2C under in cellulo conditions (Fig 4D), UBE2C overexpression only slightly restored bulk FLAG-FAT10 conjugation (Fig 6C, lane 6), suggesting a specific function of UBE2C for particular FAT10

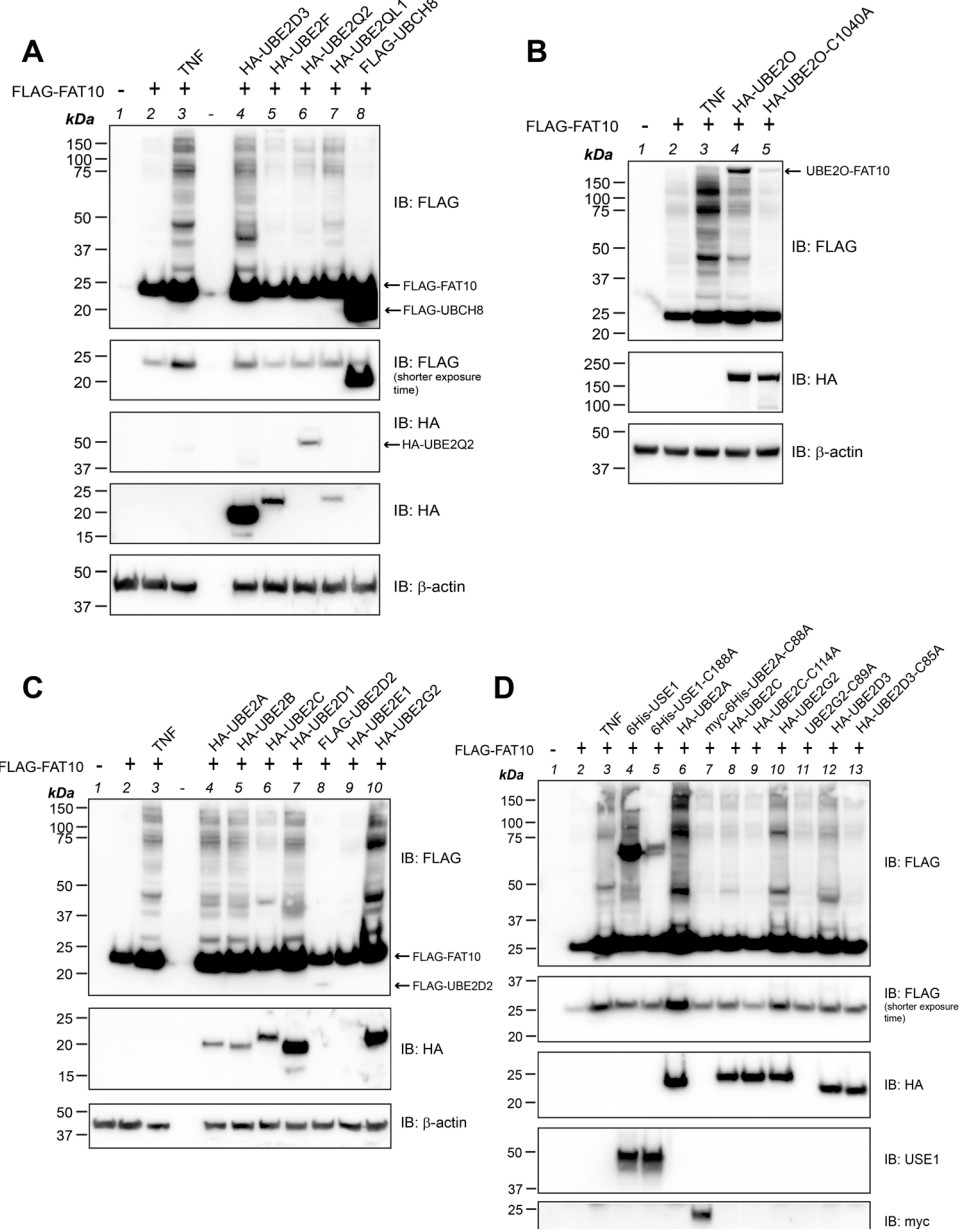

**Figure 6. Overexpression of E2 conjugating enzymes rescues FAT10 conjugation in USE1-ko cells.**

**(A, B, C, D)** HEK293-USE1-ko cells were transiently transfected with expression constructs for FLAG-FAT10 and/or the expression plasmid of the respective E2 conjugating enzymes and treated or not with TNF, as indicated. 24 h later, crude lysates were prepared under denaturing conditions and subjected to SDS–PAGE and Western blot analysis. Proteins were visualized with antibodies directed against USE1, or the HA-, myc-, or FLAG-tag of the respective proteins, and β-actin was used as a loading control. Shown is one experiment out of three experiments with similar outcomes.

Source data are available for this figure.

conjugation substrates. As a further proof for the FAT10 conjugation activity of the identified E2 conjugating enzymes, HEK293-USE1-ko cells were transiently transfected with expression plasmids for HA-UBE2A, HA-UBE2C, HA-UBE2G2, HA-UBE2D3, and UBE2O, along with their respective mutants, in which the active site cysteine was mutated to alanine, thus creating catalytically inactive E2 conjugating enzymes (Fig 6B and D). As a control, 6His-USE1 was expressed, which strongly rescued FLAG-FAT10 conjugation, whereas its active site cysteine mutant 6His-USE1-C188A was unable to do so (Fig 6D, IB: FLAG, lanes 4 and 5). The expression of HA-UBE2A, HA-UBE2G2, and HA-UBE2D3 rescued FLAG-FAT10

conjugation in almost the same manner as TNF treatment; however, their respective active site mutants were not able to perform FAT10 conjugation (Fig 6D, IB: FLAG). Because UBE2G2-C89A was expressed without a protein tag and thus its expression could not be verified, the construct pCMV-HA-UBE2G2-C89A was generated by site-directed mutagenesis of pCMV-HA-UBE2G2. The overexpression of both constructs resulted in the same outcome, namely, FAT10 conjugation in the presence of WT HA-UBE2G2 but no conjugation in the presence of its active site cysteine mutant HA-UBE2G2-C89A (Fig S4A, IB: FLAG, lanes 10, 11). Again, the overexpression of HA-UBE2C did not rescue bulk FLAG-FAT10 conjugation (Fig 6D, IB: FLAG, lane 8).

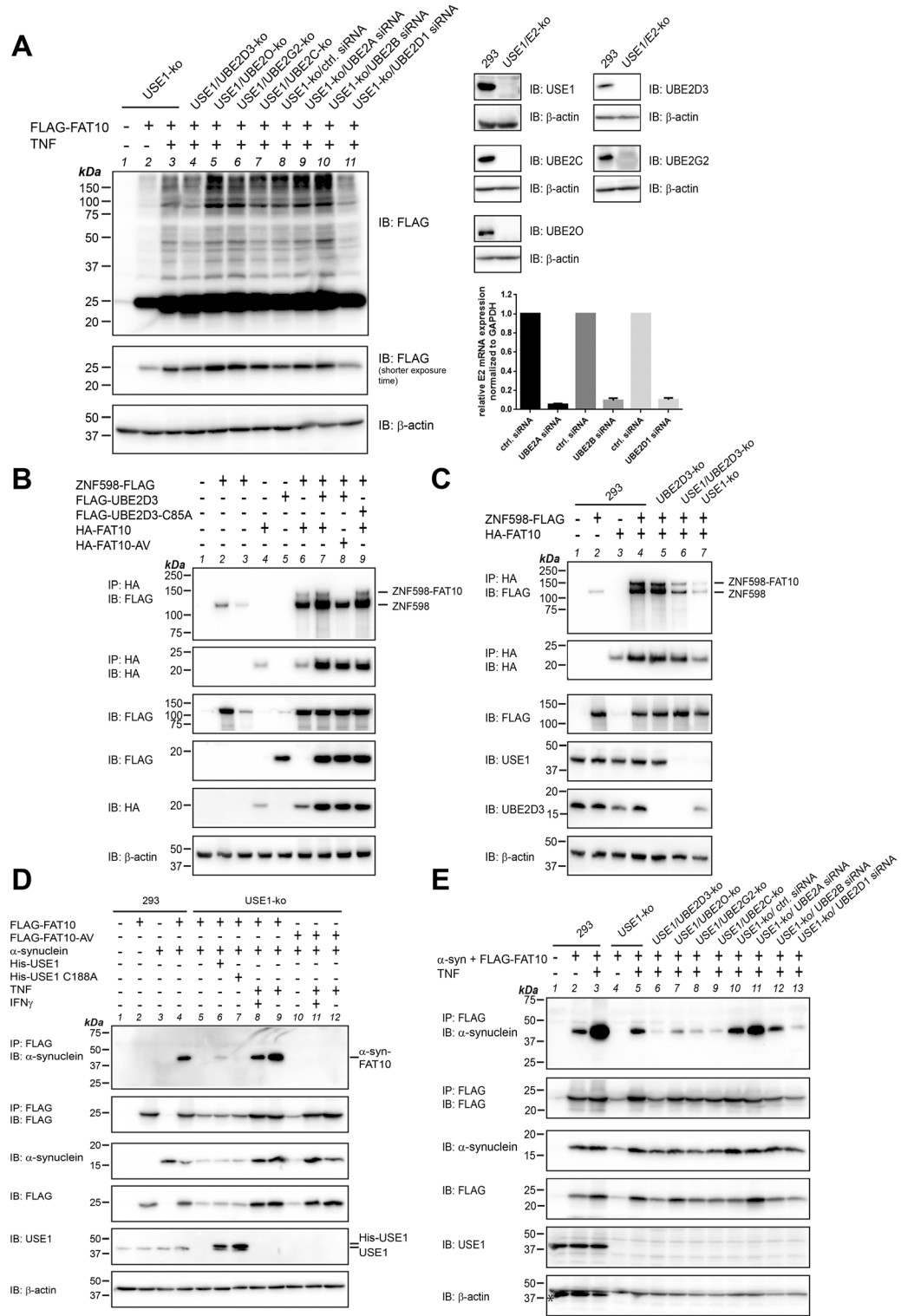

**Figure 7. USE1-independent and TNF-dependent FAT10ylation of ZNF598 and α-synuclein, respectively.**

**(A)** HEK293-USE1-ko cells (clone 01-4), double-knockout cell lines of USE1 and UBE2D3, UBE2O, UBE2G2, or UBE2C, and USE1-ko cells treated with siRNA to knock down UBE2A, UBE2B, and UBE2D1 were transfected with a FLAG-FAT10 expression plasmid and treated with TNF for 24 h, as indicated. Crude lysates were generated, and FLAG-FAT10 conjugates were analyzed on a 12.5% Laemmli gel. FLAG-FAT10 and its conjugates were visualized by Western blot analysis (IB) using a FLAG-reactive antibody, directly coupled to HRP. β-Actin was used as a loading control. Shown is one representative experiment out of three experiments with similar outcomes. Right panels show the confirmation of the double knockout by staining with E2 conjugating enzyme–reactive antibodies and β-actin as a loading control. The bar graph shows the knockdown efficiency of the respective siRNAs used for knockdown of UBE2A, UBE2B, and UBE2D1, respectively, measured by real-time PCR. Shown is the mean of three

In addition, the FAT10 E2 activity of UBE2O was likewise confirmed by comparing FLAG-FAT10 conjugation upon the overexpression of WT UBE2O or of its active site cysteine mutants UBE2O-C1040A (Fig 6B) or UBE2O-C617/C1040A (Fig S4B).

### Additional E2 conjugating enzymes are involved in the TNF-mediated rescue of FAT10 conjugation in USE1-ko cells

As demonstrated in Fig 6, the overexpression of all identified E2 conjugating enzymes, with the exception of UBE2C, could replace USE1 in FAT10 conjugation. To investigate whether one of these E2 enzymes might also be involved in the TNF-mediated FAT10 conjugation in the absence of USE1, we created, in addition to the already generated USE1/UBE2D3-ko and USE1/UBE2O-ko cell lines (Fig 3C), the double-knockout cell lines USE1/UBE2C-ko and USE1/UBE2G2-ko (Fig 7A, right panels). In case of UBE2A, UBE2B, and UBE2D1, we were not able to identify double-knockout cells, most probably because of the lack of suitable antibodies, recognizing only one single E2 isoform. Therefore, we used gene-specific siRNA to knock down UBE2A, UBE2B, and UBE2D1 in USE1-ko cells and confirmed the knockdown on the mRNA level by real-time PCR (Fig 7A, bar graph). Subsequently, the double-knockout and knockdown cell lines were tested for FLAG-FAT10 conjugation upon TNF treatment. However, it turned out that none of the E2 conjugating enzymes was involved in the TNF-mediated FAT10 conjugation because a lack or knockdown (siRNA knockdown efficiency greater than 90%) of the respective E2 enzyme in USE1-ko cells did not prevent FAT10 conjugation upon TNF treatment (Fig 7A). This result was further confirmed under endogenous conditions by performing denaturing FAT10 immunoprecipitation upon induction of endogenous FAT10 expression with IFNγ/TNF in the same E2 double-knockout (Fig S5A) and knockdown cell lines (Fig S5B). Also here, no difference in the formation of FAT10 conjugates was observed. These results suggest that besides USE1, and besides the seven newly identified E2 conjugating enzymes, at least one more E2 enzyme must exist, which is capable of performing FAT10ylation upon TNF treatment.

### The E3 ligase ZNF598 and α-synuclein can be FAT10ylated independent of USE1

To further support our finding of either USE1-independent or TNF-dependent FAT10 conjugation, we aimed to identify single substrates that are FAT10ylated in a USE1-independent manner or whose FAT10ylation depends on TNF. For this reason, we screened several of our published and yet unpublished FAT10 conjugation substrates for FAT10ylation in dependence of USE1. One of the identified FAT10 conjugation substrates was the E3 ligase ZNF598, which has been described to ubiquitylate ribosomal proteins with the help of the E2 conjugating enzyme UBE2D3 in ribosomal quality control (Garzia et al, 2017). Thus, we investigated ZNF598 FAT10ylation in dependence of both USE1 and UBE2D3. Upon expression in HEK293 WT cells, ZNF598 was FAT10ylated in a FAT10 diglycine–dependent manner (Fig 7B, IP: HA, IB: FLAG, lane 6 versus 8). However, the coexpression of UBE2D3 did not further increase the amount of the ZNF598-FAT10 conjugate (Fig 7B, IP: HA, IB: FLAG, lane 7), and the overexpression of the UBE2D3 active site cysteine mutant (FLAG-UBE2D3-C85A) showed no dominant negative effect on ZNF598 FAT10ylation (Fig 7B, lane 9). Of note, we also observed a non-covalent interaction between the two proteins (Fig 7B, IP: HA, IB: FLAG, lower band, labeled as ZNF598). Interestingly, when expressing ZNF598-FLAG and HA-FAT10 in HEK293 WT, USE1-ko, UBE2D3-ko, or USE1/UBE2D3 double-knockout cells, the ZNF598-FAT10 conjugate was always formed in all knockout cell lines (Fig 7C, IP: HA, IB: FLAG, lanes 4 versus 5–7), pointing to USE1- and UBE2D3-independent FAT10ylation of ZNF598. This result strongly supports our finding of additional E2 conjugating enzymes for FAT10ylation, besides USE1 under non-inflammatory conditions.

In addition to ZNF598, we identified a new FAT10 conjugation substrate, namely, α-synuclein, which was likewise FAT10ylated in a FAT10 diglycine–dependent manner (Fig 7D, IP: FLAG, IB: α-synuclein, lane 4 versus 10–12). In the absence of TNF, FAT10ylation of α-synuclein turned out to be dependent on USE1, because upon overexpression in HEK293 WT cells, the α-synuclein–FAT10 conjugate was formed, but not in USE1-ko cells (Fig 7D, IP: FLAG, IB: α-synuclein, lanes 4 and 5). Interestingly, α-synuclein FAT10ylation was only slightly restored in USE1-ko cells reconstituted with a USE1 expression plasmid, but not when expressing its active site cysteine mutant USE1-C188A (Fig 7D, IP: FLAG, IB: α-synuclein, lanes 6 and 7). In contrast, α-synuclein FAT10ylation was strongly restored in USE1-ko cells treated with TNF (Fig 7D, IP: FLAG, IB: α-synuclein, lane 9). Thereby, treatment with TNF alone was sufficient to mediate FAT10ylation of α-synuclein, whereas IFNγ did not further influence its FAT10ylation (Fig 7D, IP: FLAG, IB: α-synuclein, lanes 8 and 9). When HEK293 WT cells expressing α-synuclein and FLAG-FAT10 were in addition treated with TNF, the amount of the α-synuclein–FAT10

independent experiments. Control (ctrl.) siRNA–treated cells were set to unity, and the levels of the respective E2 enzymes were calculated accordingly. **(B)** HEK293 WT cells were transiently transfected with expression plasmids for ZNF598-FLAG, FLAG-UBE2D3, FLAG-UBE2D3-C85A, HA-FAT10, or HA-FAT10-AV, as indicated. After 24 h, cells were harvested and lysed in NP-40 lysis buffer. Cleared lysates were subjected to immunoprecipitation against the HA-tag of FAT10. Proteins were separated on 4–12% gradient gels (NuPAGE, Invitrogen), and ZNF598-FAT10 conjugates were visualized with a FLAG-reactive antibody, directly coupled to HRP. β-Actin was used as a loading control. Shown is one representative experiment out of three experiments with similar outcomes. **(C)** HEK293 WT (293), UBE2D3-ko, USE1/UBE2D3 double-ko, or USE1-ko cells were transiently transfected with expression plasmids for ZNF598-FLAG and HA-FAT10, as indicated. ZNF598 FAT10ylation was analyzed as described in (B). Shown is one representative experiment out of three experiments with similar outcomes. **(D)** HEK293 WT cells (293) or USE1-ko cells were transiently transfected with expression plasmids for FLAG-FAT10, FLAG-FAT10-AV, α-synuclein, His-USE1, or His-USE1-C188A, as depicted in the figure. Where indicated, cells were treated at the same time with TNF, or simultaneously with IFNγ and TNF. After 24 h, cells were harvested and lysed and cleared lysates were subjected to immunoprecipitation using FLAG-reactive antibodies. Proteins were separated on 15% Laemmli SDS–PAGE, and α-synuclein–FLAG–FAT10 conjugates were visualized with an α-synuclein–reactive antibody. The expression of all proteins was confirmed using either tag- or protein-specific antibodies, as indicated. β-Actin was used as a loading control. Shown is one representative experiment out of three experiments with similar outcomes. **(E)** HEK293 WT (293), USE1-ko, or double-knockout cell lines of USE1/UBE2D3, UBE2O, UBE2G2, or UBE2C, as well as USE1-ko cells treated with specific siRNA directed against UBE2A, UBE2B, or UBE2D1, were transiently transfected with expression plasmids for α-synuclein and FLAG-FAT10, as indicated. Formation of the α-synuclein–FLAG–FAT10 conjugate was analyzed as described in (D). An asterisk marks an unspecific stripping leftover. Shown is one representative experiment out of three experiments with similar outcomes.
Source data are available for this figure.

conjugate increased strongly as compared to conditions without TNF treatment (Fig 7E, IP: FLAG, IB: α-synuclein, lanes 2 and 3), pointing to TNF-mediated FAT10ylation of α-synuclein not only in the absence of USE1 but also in its presence. In line with our observation in Figs 7A and S5, the α-synuclein–FAT10 conjugate was still formed in all USE1/E2 double-knockout or knockdown cells, although the amount differed between the different cell lines (Fig 7E, IP: FLAG, IB: α-synuclein, lanes 5–13). This could indicate that these E2 conjugating enzymes could at least partially be involved in FAT10ylation of α-synuclein or that a knockout of the respective E2 enzyme in combination with inflammatory conditions and the overexpression of α-synuclein (a protein that is prone to aggregation) might somehow be harmful to the cells.

In summary, we have identified seven E2 conjugating enzymes, which were previously not associated with FAT10 and which are able to accept activated FAT10 from UBA6 in a FAT10 diglycine– and E2 active site cysteine–dependent manner, namely, UBE2A, UBE2B, UBE2C, UBE2D1, UBE2D3, UBE2G2, and UBE2O. With the exception of UBE2C, all identified E2 conjugating enzymes were able to reconstitute bulk FLAG-FAT10 conjugation in HEK293-USE1-ko cells, pointing to several different possible FAT10 conjugation pathways besides the so far known pathway via UBA6 and USE1. Moreover, we provide evidence that FAT10ylation can be performed by several E2 conjugating enzymes, either under conditions of constitutive FAT10 expression as, for example, in certain immune cells or under inflammatory conditions in the presence of TNF. In contrast to our initial hypothesis of an E2 enzyme whose mRNA expression might be induced by TNF, we rather suggest that the E2 conjugating enzyme(s) become(s) activated by a TNF-mediated process, which still needs to be explored in future experiments.

## Discussion

According to the current paradigm, USE1 is regarded to be the major and maybe only E2 conjugating enzyme for the conjugation of the ULM FAT10. In the current study, we provide striking evidence that besides USE1, at least seven additional ubiquitin E2 conjugating enzymes show activity as E2 conjugating enzymes for FAT10ylation. Interestingly, most of these E2 conjugating enzymes such as UBE2A, UBE2B, UBE2D1, UBE2D3, UBE2O, or UBE2G2 were able to restore broadly visible, overall FAT10 conjugation in USE1-ko cells. In contrast, UBE2C was unable to do so, although active site cysteine– and/or FAT10 diglycine–dependent loading of FAT10 onto UBE2C was clearly visible. This suggests that UBE2C might be specific for only few FAT10 conjugation substrates, whereas the other identified E2 enzymes might have a broader function in FAT10 conjugation. Alternatively, the cellular environment in HEK293 cells might not be suitable for UBE2C-mediated bulk FAT10 conjugation, for example, because of missing substrates, E3 ligases, or other factors.

While UBE2A, UBE2B, UBE2D1, UBE2D3, and UBE2G2 belong to the E2 enzyme class I, which contains the UBC domain only, USE1 and UBE2O belong to class IV and are characterized by additional N- and C-terminal extensions (van Wijk & Timmers, 2010). UBE2C, however, contains an N-terminal extension and, thus, belongs to class II, similar to UBE2E1, which was loaded with FAT10 in a FAT10

diglycine–dependent manner, but which could not further be confirmed as an E2 enzyme for FAT10 because of low expression levels (Figs 5 and 6). Unfortunately, it is still unknown how the different classes of E2 enzymes interact with UBA6 and no crystal structure for UBA6 interacting with an E2 enzyme is available yet. Thus, it cannot be clarified how UBA6 selects the respective E2 conjugating enzyme for FAT10 transfer. It will be interesting to investigate the specific affinities of UBA6 with USE1 as compared to the herein-identified FAT10-reactive E2 conjugating enzymes in future experiments to shed light on the regulation of FAT10 conjugation by different E2 enzymes.

FAT10 conjugation in USE1-ko cells was rescued upon treatment with TNF, pointing to one or more FAT10-specific E2 conjugating enzymes, which might either be up-regulated upon TNF treatment on their mRNA level or be activated by a TNF-dependent mechanism. Of the six E2 conjugating enzymes that were tested based on our transcriptome analysis, UBE2D3 and UBE2O were confirmed by in vitro and in cellulo experiments as E2 conjugating enzymes for FAT10. With the exception of UBE2O and UBE2Q2, the mRNAs of all other E2 enzymes were significantly up-regulated upon TNF treatment, as compared to their levels in untreated HEK293 cells. However, the degree of up-regulation was neglectable and found to be in a range of only 1.1–2.5-fold. Therefore, we suggest that the activation of the E2 conjugating enzyme(s) might rather take place on the posttranslational level. Indeed, E2 conjugating enzymes were recently divided into two groups, namely, constitutively active and regulated E2 enzymes dependent on the presence of either an aspartic acid (Asp120, constitutively active) or a phosphorylatable serine (Ser120, regulated) in the immediate proximity of the active site cysteine pocket (Valimberti et al, 2015). Interestingly, although UBE2A, UBE2B, UBE2C, and UBE2G2 bear a serine (Ser) at this site and thus can be classified as regulated E2 conjugating enzymes, UBE2D1 and UBE2D3 possess an aspartic acid (Asp) at this position, classifying them as constitutively active enzymes. USE1 and UBE2O differ from this scheme and contain a glutamic acid (Glu) at the same position. Glutamic acid is often used by default to create phosphomimetic variants of proteins by site-directed mutagenesis. This suggests that USE1 and UBE2O rather belong to the group of constitutively active E2 conjugating enzymes. This finding might provide an explanation for why USE1 seems to have such dominant activity for FAT10ylation, given that a USE1 knockout almost completely abrogated FAT10 conjugation in the absence of TNF (Fig 1A). Within this context, it is interesting to note that phosphorylation of UBE2A by cyclin-dependent kinase-9 (CDK9) was described to activate the UBE2A's E2 conjugating enzyme activity, whereas CDK9 knockdown significantly impaired its activity with respect to monoubiquitylation of the proliferating cell nuclear antigen (Shchebet et al, 2012). In our experiments, UBE2A was loaded with FAT10 onto its active site cysteine (Figs 4A and B and S3) and the overexpression of UBE2A, but not of its active site cysteine mutant, rescued conjugation of FLAG-FAT10 in HEK293-USE1-ko cells (Fig 6). This finding makes UBE2A an interesting candidate being capable of restoring FAT10 conjugation in TNF-treated USE1-ko cells. However, upon siRNA-mediated knockdown of UBE2A mRNA in USE1-ko cells, TNF-mediated FLAG-FAT10 conjugation, as well as endogenous FAT10 conjugation, was unaffected (Figs 7A and S5B). The same result was obtained for all other, in this study, identified E2

conjugating enzymes (Figs 7A and S5A and B), making them all unlikely to be the E2 conjugating enzyme, which might be involved in TNF-dependent FAT10ylation. Moreover, these results point to the existence of at least one additional E2 conjugating enzyme, that is activated by TNF and has yet to be identified. The binding of TNF to cell surface receptors activates several signal transduction pathways, leading to the activation of several types of kinases such as MAPK, JNKs, or extracellular signal–regulated kinases (Wajant et al, 2003; Sabio & Davis, 2014). Thus, it is very likely that certain E2 conjugating enzymes are activated by TNF-mediated phosphorylation, making them competent for FAT10 conjugation. However, we also do not want to exclude the possibility that the TNF-mediated FAT10ylation might eventually be mediated not by a TNF-activated E2 enzyme, but maybe by another factor. For example, this could be an E3 ligase, which could be activated by TNF, rendering it capable to interact with a FAT10-specific E2 enzyme, other than USE1.

Apart from the fact that none of the seven identified E2 conjugating enzymes turned out to be the one, activated by TNF treatment, we nevertheless confirmed that all could be loaded with FAT10 in a FAT10 diglycine– and E2 active site cysteine–dependent manner, either in the absence or in the presence of USE1. Important questions that arise from this finding are about the need for additional E2 conjugating enzymes besides USE1, and furthermore, which E2 conjugating enzyme is active for FAT10 conjugation under which conditions, in which cell type, and for which particular conjugation substrate? Screening of FAT10 interaction partners and covalent conjugation substrates revealed that FAT10 gets conjugated to hundreds of proteins (Aichem et al, 2012); thus, regulation on the level of the E2 conjugating enzymes in combination with different E3 ligases might be a possibility, although it is entirely unclear how this could be regulated. Upon screening of several of our identified FAT10 conjugation substrates, we identified the E3 ligase ZNF598 as a USE1-independent FAT10ylation substrate (Fig 7B and C), whereas α-synuclein FAT10ylation was dependent on USE1 under overexpressing, non-inflammatory conditions, but strongly FAT10ylated upon TNF treatment, either in USE1-ko or in HEK293 WT cells (Fig 7D and E). Interestingly, also FAT10ylation of the two E2 conjugating enzymes UBE2Q2 and UBE2QL1 was detectable in USE1-ko cells only upon treatment of the cells with TNF (Fig S2C and D). These results support our finding of additional E2 conjugating enzymes besides USE1 for FAT10 conjugation. Moreover, they in addition support our hypothesis of the existence of at least one additional TNF-dependent E2 enzyme. However, it also indicates complicated regulation of the usage of the different E2 enzymes for FAT10ylation. Cells normally do not express FAT10 in the absence of inflammatory triggers such as IFNγ and TNF. Based on our results, USE1 apparently does not play a prominent role under inflammatory conditions because FAT10 conjugation could easily be restored in USE1-ko cells upon exposure to TNF. In contrast, a high basal expression of FAT10 was observed in organs of the immune system such as lymph node, spleen, or thymus (Lee et al, 2003; Canaan et al, 2006; Lukasiak et al, 2008) where FAT10 expression was assigned to specific immune cells such as mature dendritic cells (DCs), B cells, and medullary thymic epithelial cells (so-called mTECs). Likewise, FAT10 mRNA was detectable in human CD8[+] T cells, natural killer cells (NK cells), and natural killer T cells (NKT cells) under constitutive conditions (Bates et al, 1997; Buerger

et al, 2015; Schregle et al, 2018). Based on these findings, we suggest that FAT10 might have cell type–specific functions and one possibility to regulate this might be the usage of the FAT10 E2 conjugating enzymes described in this study. In order to prove this hypothesis, it will be interesting to investigate in future experiments whether and how the E2 conjugating enzymes identified in this study are regulated and whether they are expressed in a cell type–specific manner. Altogether, these findings will help to better understand the broad function of FAT10 not only under different disease conditions such as cancer or viral infection but also under constitutive conditions when expressed in specific immune cells.

## Materials and Methods

### Cell culture and generation of CRISPR/Cas9-based knockout cell lines

The human embryonic kidney cell line HEK293 was cultivated in HyClone IMDM (VWR International GmbH), supplemented with 10% FCS (Gibco/Thermo Fisher Scientific), 1% stable glutamine (100x, 200 mM), and 1% penicillin/streptomycin (100x) (both from Biowest/VWR). The human hepatocellular carcinoma cell line Huh7 was cultivated in HyClone DMEM (VWR International GmbH), supplemented with 10% FCS (Gibco/Thermo Fisher Scientific) and 1% penicillin/streptomycin (100x). The human colon cancer cell line HCT116 was cultivated in HyClone Medium *RPMI 1640* (VWR International GmbH), supplemented with 10% FCS (Gibco/Thermo Fisher Scientific) and 1% penicillin/streptomycin (100x) (Biowest/VWR). All cell lines were regularly tested to be negative for Mycoplasma infection using the MycoAlert kit (Roche).

The generation of the CRISPR/Cas9-based knockout cell lines HEK293-USE1-ko, HEK293-FAT10-ko, and HEK293-UBA6-ko has been described before (Aichem et al, 2018, 2019a, 2019b). HEK293-UBA6/USE1 double-ko cells were generated by transfection of HEK293-USE1-ko cells (clone 01-4) with a pCMV-Cas9-GFP plasmid, containing UBA6-specific gRNA Hs0000418715 (Sigma-Aldrich [Aichem et al, 2019b]). HEK293-UBE2D3-ko and HEK293-USE1/UBE2D3-ko cells were generated using the KN2.0 non-homology–mediated CRISPR gene knockout kit (OriGene) with UBE2D3-specific gRNA, as described by the manufacturer. HEK293-UBE2O-ko cells were generated using the UBE2O-specific gRNA Hs0000452881 inserted into the plasmid pCMV-Cas9-GFP (Sigma-Aldrich). The same plasmid was used for the generation of HEK293-USE1/UBE2O and HEK293-UBA6/USE1/UBE2O triple-ko cells by transfecting either HEK293-USE1-ko cells or HEK293-UBA6/USE1 double-ko cells with a pCMV-Cas9-GFP-UBE2O gRNA plasmid, respectively. Huh7-USE1-ko and HCT116-USE1-ko were generated in the same way, using the USE1-specific gRNA Hs0000460901 (Sigma-Aldrich [Aichem et al, 2018]). 24 h after transfection, GFP[high] cells were sorted using BD FACSAria IIu (BD Biosciences). Single-cell clones were cultivated in the respective cell-specific medium. Western blot analysis was used to confirm the successful knockout using a USE1-reactive polyclonal antibody (Aichem et al, 2010), a rabbit polyclonal UBA6-reactive antibody (Enzo Life Sciences [Aichem et al, 2010]),

a UBE2O-reactive antibody (anti-UBE2O rabbit polyclonal antibody, ab254592; Abcam, 1:250), or a UBE2D3-reactive antibody (SAB2102622; Sigma-Aldrich, 1:1,000). USE1/UBE2C double-ko and USE1/UBE2G2 double-ko cell lines were generated by transfection of HEK293-USE1-ko cells (clone 01-4) with pCMV-Cas9-GFP containing gRNAs directed against UBE2C (Hs0000341305) and UBE2G2 (Hs0000195749). Western blot analysis was used to confirm the successful knockout using UBE2C-reactive (ab252940; Abcam, 1:1,000) and UBE2G2-reactive (WH0007327M1; Sigma-Aldrich, 1:420) antibodies.

## Endogenous FAT10 expression and transient transfection of cells

For the induction of endogenous FAT10 expression, cells were seeded with a density of $1–2 × 10^6$ cells/cell culture dish (10 cm). After 24 h, the expression of endogenous FAT10 was induced by the addition of 300 U/ml IFNγ and 600 U/ml TNFα (both from Pepro-Tech), as described in Aichem et al (2019a). For transient transfection, cells were seeded as described above. After 24 h, cells were transfected by lipofection using 19.2 $\mu$l of TransIT-LT1 transfection reagent (Mirus) and 6.6 $\mu$g of total plasmid/cell culture dish (10 cm), as described by the manufacturer. Cells were incubated for at least 24 h at 37°C and 5% $CO_2$ before harvesting.

## Cell lysates, immunoprecipitation, and Western blot analysis

To monitor in cellulo FAT10 loading onto E2 conjugating enzymes, cells were harvested and lysed as follows: after removal of the cell culture medium, cells were washed once with PBS, subsequently lysed for 30 min in 1–1.3 ml per 10-cm cell culture dish NP-40 lysis buffer (20 mM Tris–HCl, pH 7.6, 50 mM NaCl, 10 mM $MgCl_2$, and 1% NP-40), supplemented with 1x protease inhibitor mix (Mini; cOmplete, EDTA-free Protease Inhibitor Cocktail Tablets [Roche]), and subsequently centrifuged at 4°C and 20,000$g$ for 30 min. Cleared lysates were subjected to immunoprecipitation using HA-reactive, agarose-coupled antibodies (clone HA-7) (Sigma-Aldrich/Merck). Beads were washed twice with 1 ml of NET-TN wash buffer (50 mM Tris–HCl, pH 8.0, 650 mM NaCl, 5 mM EDTA, and 0.5% Triton X-100) and twice with NET-T wash buffer (50 mM Tris–HCl, pH 8.0, 150 mM NaCl, 5 mM EDTA, and 0.5% Triton X-100), and boiled in a standard 5x SDS gel sample buffer supplemented with (reducing conditions) or without (non-reducing conditions) 4% $β$-2-ME. Crude lysates were generated as described before (Aichem et al, 2019b). Briefly, cells were washed once with PBS, supplemented with 10 mM N-ethyl-maleimide (NEM). 400–500 $\mu$l of 5x SDS gel sample buffer, supplemented with 10 mM NEM, was directly added to the cell culture dish (10 cm), and lysates were transferred into 1.5-ml reaction tubes, subsequently sonified for 20 s, and boiled for 4 min. Before loading onto SDS–PAGE, samples were centrifuged at 20,000$g$ at RT for 1 min at maximum speed. For immunoprecipitation under denaturing conditions (as described earlier [Aichem et al, 2018]), cells were washed once with PBS/10 mM NEM and lysed directly in the cell culture dish (10 cm) in 250 $\mu$l SDS lysis buffer (1x PBS, 2% SDS, 10 mM EDTA, pH 8.0, 10 mM EGTA, pH 8.0, and 10 mM NEM) and 1x protease inhibitor mix (Mini; cOmplete, EDTA-free Protease Inhibitor Cocktail Tablets [Roche]). Lysates were collected in 1.5-ml reaction tubes and sonified twice for 20 s each. After the addition of

50 $\mu$l 1M DTT, samples were boiled for 10 min. Renaturation was performed on ice by diluting the boiled samples in 10 volumes of RIPA buffer (50 mM Tris–HCl, pH 8.0, 150 mM NaCl, 1% Triton X-100, 0.5% Na-deoxycholate, and 0.1% SDS), supplemented with 10 mM EDTA, pH 8.0, 10 mM EGTA, pH 8.0, 10 mM NEM, and 1x protease inhibitor mix (Mini; cOmplete, EDTA-free Protease Inhibitor Cocktail Tablets [Roche]). Renatured samples were filtered (0.45 $\mu$m) and subjected to immunoprecipitation using anti-FLAG M2 affinity gel (Sigma-Aldrich) or a FAT10-reactive mouse monoclonal antibody (clone 4F1) (Enzo Life Sciences [Aichem et al, 2010]) bound to protein A Sepharose. Beads were washed twice with NET-TN and NET-T buffer, as described above, supplemented with 5x SDS gel sample buffer supplemented with 4% 2-ME, and boiled.

Proteins were separated on either standard 12.5% Laemmli SDS gels, 4–12% NuPAGE (Invitrogen), or 4–12% mPAGE (Sigma-Aldrich) gradient gels and subjected to Western blot analysis. Endogenous FAT10 was detected using a rabbit polyclonal FAT10-reactive antibody (1:1,000; Enzo Life Sciences [Hipp et al, 2004]), USE1 was detected using a USE1-reactive rabbit polyclonal antibody (1:1,000; Enzo Life Sciences [Aichem et al, 2010]), and UBA6 was detected using an anti-UBA6 rabbit polyclonal antibody (1:1,000; Enzo Life Sciences [Aichem et al, 2010]). Anti-UBE2D3 rabbit polyclonal antibody (SAB2102622; Sigma-Aldrich, 1:1,000) and anti-UBE2O rabbit polyclonal antibody (ab254592; Abcam, 1:250) were used for visualization of endogenous UBE2D3 and UBE2O, respectively. Anti-UBE2C (ab252940; Abcam, 1:1,000), and anti-UBE2G2 (WH0007327M1; Sigma-Aldrich, 1:420) antibodies were used to detect endogenous UBE2C and UBE2G2, respectively. An $α$-synuclein–reactive antibody (ab138501; Abcam, 1:10,000) was used to detect endogenous and overexpressed $α$-synuclein. Directly peroxidase-coupled antibodies anti-FLAG-HRP (clone M2; Sigma-Aldrich, 1:3,000) and anti-HA-HRP (clone HA-7; Sigma-Aldrich, 1:4,000) were used to detect FLAG- and HA-tagged proteins, respectively. A monoclonal $β$-actin–reactive antibody (ab6276; Abcam, 1:5,000; clone AC-15) was used as a loading control.

## Plasmids and primers

Plasmids used for transient transfection of human HEK293, Huh7, or HCT116 cells with FLAG-tagged FAT10 variants were pcDNA3.1-His-3xFLAG-FAT10 (Jin et al, 2007) for the expression of 6His-3xFLAG-FAT10 (FLAG-FAT10), and pcDNA3.1-His-3xFLAG-FAT10-AV (Aichem et al, 2010) for the expression of the conjugation-incompetent FAT10 diglycine mutant FLAG-FAT10-AV. pcDNA3.1-HA-FAT10 and pcDNA3.1. HA-FAT10-AV have been published in Hipp et al (2004) and Aichem et al (2010). Plasmids pcDNA-3.1-His/-A-USE1 and its active site mutant pcDNA-3.1-His/-A-USE1-C188A have been described earlier in Aichem et al (2010). Plasmids for the recombinant expression of His-tagged E2 conjugating enzymes and the generation of plasmids for the expression of HA-tagged E2 conjugating enzymes in human cell culture are listed in Tables S1 and S2. Shortly, pCMV-HA-E2 expression plasmids were generated by PCR amplification using the respective oligonucleotides listed in Table S1 and the respective recombinant E2 expression plasmids listed in Table S3 as a template. Amplicons were inserted into an EcoRI/XhoI or BglII/KpnII-digested plasmid pCMV-HA (Clontech), creating an N-terminal HA-tag to the respective E2 conjugating enzyme. All sequences were verified by sequencing (Microsynth AG). Active site cysteine

mutants were generated by a standard site-directed mutagenesis protocol using the respective oligonucleotides listed in Table S2.

FLAG-tagged UBE2O truncation variant FLAG-UBE2O trunc (nt2,434–3,879) containing the coiled-coil domain, the UBC domain, and the complete C-terminus of UBE2O was generated by PCR amplification using oligonucleotides AA-450 5′CCGGAATTCGAGATAGAACCCGGGAGTT-GAAAGAGGCCATCAAG-3′ and AA-451 5′GCCGGAGTGCACAGAGGACAAGT-AGTCTAGACTAG-3′, and inserted into an EcoRI/XbaI-digested plasmid pcDNA3.1-3xFLAG-TEV-UBE2O (see Table S1) to replace full-length UBE2O. Active site cysteine mutants of UBE2O (C1040A, C617A, and C1040/C617A) were generated by site-directed mutagenesis using the primers listed in Table S2.

pcDNA3.1-HA-TEV-UBE2O and its active site cysteine mutants pcDNA3.1-HA-TEV-UBE2O-C1040A and pcDNA3.1-HA-TEV-UBE2O-C617/1040A were generated by PCR cloning using primers PR6-125′-EcoRI-UBE2O 5′-CCGGAATTCTCATGGCGGATCCCGCAGCCCCCACG-3′ and PR6-133′-NotI-UBE2O 5′-TTTTCCTTTTGCGGCCGCCTACTTGTCCTCTGTG-CACTCCGGCATGCCTG-3′ with pcDNA3.1-3xFLAG-TEV-UBE2O, pcDNA3.1-3xFLAG-TEV-UBE2O-C1040A, or pcDNA3.1-3xFLAG-TEV-UBE2O-C617/1040A as a template, respectively. To generate the expression plasmid for HA-tagged FAT10 C0 (C134L), site-directed mutagenesis was performed using pTYB-HA-FAT10(-1aa)-intein-CBD (chitin binding domain) (kindly provided by Dr. Benedict Kessler [Hemelaar et al, 2004]). The following primers were used as single oligonucleotides to mutate all cysteine residues: 5′-CTCCCAATGCTTCCAGCCTCTCTGTG-CATGTCCGTT-3′, 5′-TCCCTGAGACCCAGATTGTGACTAGCAATGGAAAGA-3′, and 5′-CTCTTCCTGGCATCTTATTCTATTGGATGCTTTGCC-3′. To generate the C134L mutation, the following primer pair was used: 5′-CATCTTCCAGTCTCTTTCCATTCAGAGTCACAATCTGGGTTTCAGGG-3′ and 5′-CCCTGAAACCCAGATTGTGACTCTGAATGGAAAGAGACTGGAAGATG-3′. To achieve 6His-pSUMO-HA-FAT10 C0 (C134L), a PCR- and restriction digest–based transfer of HA-FAT10 C0 (C134L) from pTYB2-HA-FAT10 C0 (C134L) was performed using the following primers and restriction enzymes, respectively: forward primer 5′-CCAGTGCCTCTCAGGTGGT-TACCCCTACGACGTGCCCGAC-3′ (isoschizomer for BsaI: BsmBI), and reverse primer 5′-GTGCTCGAGCTAACATCCAATAGAATAAGATG-3′ (XhoI; both from FastDigest; Thermo Fisher Scientific). pSUMO-HA-FAT10 C0 (C134L) GA was generated with primers 5′-CCTGGCATCTTATTCTATTG-GAGCTTAGCTCGAGCACCACCC-3′ and 5′-GGGTGGTGCTCGAGCTAAGCTC-CAATAGAATAAGATGCCAGG-3′. pSUMO-HA-FAT10 C0 (C134L)-GG was generated with primers 5′-CCTGGCATCTTATTCTATTGGAGGT-TAGCTCGAGCACCAC-3′ and 5′-GTGGTGCTCGAGCTAACCTCCAATA-GAATAAGATGCCAGG-3′. pcDNA6 asyn WT (107425; Addgene plasmid [Mbefo et al, 2015]) was used to express α-synuclein, and pcDNA3.1-ZNF598-TEV-3xFLAG (105690; Addgene plasmid [Juszkiewicz & Hegde, 2017]) was used for the expression of ZNF598-3xFLAG.

## Purification of recombinant 6His-tagged E2 conjugating enzymes

For recombinant protein purification, plasmids with pDEST17 or pET15 vector backbone (see Table S3) were transformed into chemically competent *E. coli* BL21 (DE3) (Novagen) by heat shock. All proteins were expressed in modified LB medium (13.5 g/Liter peptone, 7 g/Liter yeast extract, 14.9 g/Liter glycerol, 2.5 g/Liter NaCl, 2.3 g/Liter $K_2HPO_4$, 1.5 g/Liter $KH_2PO_4$, and 0.14 g/Liter $MgSO_4 \times 7 H_2O$ [pH 7.0]). After overnight growth, cultures were diluted in fresh medium to an $A_{600}$ of 0.1 and cultivated until an $A_{600}$ of 0.5–0.7 was

reached. Recombinant protein expression was induced at 21°C overnight with a final concentration of 0.4 mM IPTG. Cells were harvested by centrifugation (3,220$g$, 30 min, 4°C) and lysed in bacterial lysis buffer (50 mM Tris–HCl, pH 7.5, 10% glycerol, 0.1% Triton X-100, 100 $\mu$g/$\mu$l lysozyme, and 1M PMSF) and 1x protease inhibitor mix (Mini; cOmplete, EDTA-free Protease Inhibitor Cocktail Tablets [Roche]) (10 ml/Liter g pellet). Lysates were cleared by centrifugation (32,000$g$, 8°C, 30 min), and 500 $\mu$l $Ni^{2+}$-NTA-agarose (QIAGEN) pre-equilibrated in lysis buffer was added to each lysate and incubated for 1 h at 4°C with rolling. Beads were collected by centrifugation (2,000$g$, 4°C, 1 min) and washed twice with binding buffer (20 mM Tris–HCl, pH 7.5, 500 mM NaCl, 20 mM imidazole, and 1 mM TCEP). Elution of 6His-tagged proteins was performed by the addition of 500 $\mu$l elution buffer (20 mM Tris–HCl, pH 7.5, 500 mM NaCl, 500 mM imidazole, and 1 mM TCEP). The elution buffer was finally exchanged by storage buffer (20 mM Tris–HCl, pH 7.5, 150 mM NaCl, and 0.2% TCEP), using PD MiniTrap G-25 columns and the spin protocol after the manufacturer's instructions (GE Healthcare). For long-time storage at –80°C, protein eluates were supplemented with 10% glycerol.

## Recombinant proteins and in vitro assays

Recombinant 6His-tagged E2 conjugating enzymes were purified as described above. The purification of 6His-tagged USE1 has been described earlier (Aichem et al, 2010; Boehm et al, 2020), and purification of FAT10 and its conjugation-incompetent mutant FAT10-AV was described earlier (Aichem et al, 2019a). HA-tagged stabilized FAT10 HA-FAT10 C0 (C134L)-GG or its conjugation-incompetent variant HA-FAT10 C0 (C134L)-GC was purified in the same way as described earlier for FAT10 (Aichem et al, 2019a). Detailed purification protocols can be obtained upon request. FLAG-UBA6 and 6His-ubiquitin were purchased from Enzo Life Sciences. To investigate FAT10 activation by UBA6 and its subsequent loading onto the different E2 conjugating enzymes, 0.2 $\mu$g FLAG-UBA6, 5 $\mu$g FAT10, 5 $\mu$g FAT10-AV, or 2 $\mu$g 6His-ubiquitin was mixed in 20 $\mu$l 1x in vitro buffer (20 mM Tris–HCl, pH 7.6, 50 mM NaCl, 10 mM $MgCl_2$, 4 mM ATP, and 0.1 mM DTT) in the presence of either 2 $\mu$g 6His-USE1 or 5 $\mu$l of the different E2 conjugating enzymes, and incubated for 30 min at 37°C. Reactions were stopped by the addition of 5x SDS gel sample buffer with (reducing) or without 4% 2-ME (non-reducing), and boiling.

FLAG-tagged UBE2O variants were purified from HEK293-UBA6/USE1/UBE2O triple-ko cells. 24 h after transient transfection with the respective UBE2O expression plasmids, cells were lysed by standard NP-40 lysis (as described above) and cleared lysates were subjected to immunoprecipitation using anti-FLAG M2 affinity gel (Sigma-Aldrich). Beads were washed three times with NET-TN wash buffer, three times with NET-T wash buffer as described above, and one time with 1x in vitro buffer. The in vitro reaction was performed directly on the beads. Reactions were stopped by the addition of 5x SDS gel sample buffer with (reducing) or without 4% 2-ME (non-reducing), and boiling.

In vitro activation and loading of ISG15 or FAT10 onto the E2 conjugating enzyme UBE2L6 (UBCH8) were performed using recombinant 6His-tagged ISG15 (1 $\mu$g/$\mu$l), 6His-UBE1L (1 $\mu$g/$\mu$l), 6His-UBE2L6 (UBCH8) (0.5 $\mu$g/$\mu$l), and FLAG-UBA6 (0.58 $\mu$g/$\mu$l) (all

from Enzo Life Sciences). Reactions were performed in a volume of 20 $\mu$l with the following molar protein concentrations: 6His-ISG15 (2.6 $\mu$M), 6His-UBE1L (440 nM), and 6His-UBE2L6 (690 nM); FLAG-UBA6 (120 nM); and FAT10/FAT10-AV (13 $\mu$M). Reactions were performed in 1x in vitro buffer as described above.

### Transcriptome analysis

HEK293 WT cells were treated or not for 24 h with 600 U/ml TNF. Total mRNA was isolated using the RNeasy Mini kit (QIAGEN). Samples were analyzed at the Functional Genomics Center Zurich for mRNAs, which were significantly up-regulated as compared to the untreated control. In brief, RNA integrity was determined using the fragment analyzer (Agilent Technologies). The RIN of the samples submitted for library preparation ranged from 8.4 to 9.2. Sequencing libraries were prepared according to the manufacturer's instructions of the TruSeq Stranded mRNA library prep kit (Illumina) using 300 ng input of total RNA per sample. Quality-controlled libraries were sequenced in one multiplex on a NovaSeq instrument (Illumina) generating ~40 Mio reads per sample (100 bp single-end). Quality-based adapter trimming was executed with Trimmomatic. Filtered and trimmed sequences were aligned to the human genome build GRCh38.p13 from ENSEMBL using the STAR aligner (Dobin et al, 2013). Raw read counts were calculated with featureCounts of the Rsubread package (Liao et al, 2013) and proceeded to differential analysis and read count normalization using edgeR (Robinson et al, 2010). All analyses were run by the SUSHI platform (Hatakeyama et al, 2016) of the Functional Genomics Center Zurich. The complete list of hits is available in Table S4.

### siRNA transfection and real-time PCR

HEK293-USE1-ko cells (clone 01-4) were transfected with 40 nM in total of a pool of four different siRNAs directed against human UBE2A mRNA (SI02663745, SI3108665, SI03079713, and SI00050995, FlexiTube siRNA; QIAGEN), UBE2B mRNA (SI000051002, SI02630922, SI03063277, and SI03103863, FlexiTube siRNA; QIAGEN), or UBE2D1 mRNA (SI00302085, SI04290048, SI04226026, and SI0415212, FlexiTube siRNA; QIAGEN), or with 40 nM unspecific control siRNA (AllStars Negative Control siRNA; QIAGEN) using HiPerFect transfection reagent (QIAGEN), as described by the manufacturer. 1 d later, cells were transfected with pcDNA3.1-His-3xFLAG-FAT10 (Chiu et al, 2007) using TransIT-LT1 transfection reagent (Mirus) and incubated for an additional 24 h. Cells were lysed either by generating crude lysates or by performing denaturing immunoprecipitation using the FAT10-reactive monoclonal antibody 4F1 (Enzo Life Sciences [Aichem et al, 2010]), as described above. Knockdown of UBE2A, UBE2B, and UBE2D1 on the mRNA level was verified by real-time PCR. RNA was isolated using the RNeasy kit (QIAGEN), and cDNA was subsequently reverse-transcribed using High-Capacity cDNA Reverse Transcription Kit (Applied Biosystems/Life Technologies). Real-time PCR was performed on Applied Biosystems 7900-HT Fast Real-Time PCR Cycler using Fast SYBR Green Master Mix (Applied Biosystems/Life Technologies) with gene-specific primers (QuantiTect Primer Assays; QIAGEN) for *ube2a* (Hs_UBE2A_1_SG), *ube2b* (Hs_U-BE2B_1_SG), and *ube2d1* (Hs_UBE2D1_1_SG). *gapdh* (Hs_GAPDH_1_SG) served as a housekeeping gene.

## Data Availability

This study includes no data deposited in external repositories. Original, uncropped, and unprocessed scans of all gels used in the figures are shown in Source Data. Supplementary data for this article are available.

## Supplementary Information

## Acknowledgements

We gratefully acknowledge deceased Prof. Dr. Marcus Groettrup for scientific support and discussions and for acquiring funding resources. We thank Dr. Edith Uetz van Allmen and Ilona Kindinger for technical help, Hubert Rehrauer from the Functional Genomics Center of the ETH Zuerich for the helpful discussion of the transcriptomics data, and PD Dr. Klaus-Peter Knobeloch and Dr. Stefanie Mueller for critical reading of the article. This work was supported by the Swiss State Secretariat for Education, Research and Innovation, Swiss Velux Foundation (projects 855 and 1029), the DFG Collaborative Research Center CRC969 (TP C01 and C09), and the Graduate School KoRS-CB (to J Bialas) at the University of Konstanz in Germany.

### Author Contributions

L Schnell: investigation.
A Zubrod: investigation.
N Catone: investigation.
J Bialas: investigation.
A Aichem: conceptualization, data curation, formal analysis, supervision, funding acquisition, validation, investigation, methodology, project administration, and writing—original draft, review, and editing.

### Conflict of Interest Statement

The authors declare that they have no conflict of interest.

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
