## [Reviewer comments · Life Science Alliance]

Life Science Alliance

Tumor necrosis factor mediates USE1-independent FAT10ylation under inflammatory conditions

Leonie Schnell, Alina Zubrod, Nicola Catone, Johanna Bialas, and Annette Aichem

DOI: <https://doi.org/10.26508/lsa.202301985>

Corresponding author(s): Annette Aichem, Biotechnology Institute Thurgau

Review Timeline:

Submission Date:	2023-02-10
Editorial Decision:	2023-03-20
Revision Received:	2023-06-19
Editorial Decision:	2023-08-10
Revision Received:	2023-08-11
Accepted:	2023-08-11

Transaction Report:

March 20, 2023

Re: Life Science Alliance manuscript #LSA-2023-01985-T

Dr. Annette Aichem
Biotechnology Institute Thurgau
Unterseestrasse 47
Kreuzlingen 8280
Switzerland

Dear Dr. Aichem,

Thank you for submitting your manuscript entitled "Newly identified E2 enzymes for FAT10ylation reveal USE1-independent conjugation in inflammation" to Life Science Alliance. The manuscript was assessed by expert reviewers, whose comments are appended to this letter. We invite you to submit a revised manuscript addressing the Reviewer comments.

Thank you for this interesting contribution to Life Science Alliance. We are looking forward to receiving your revised manuscript.

Sincerely,

B. MANUSCRIPT ORGANIZATION AND FORMATTING:

Reviewer #1 (Comments to the Authors (Required)):

In this manuscript, Schnell et al. identify E2 enzymes that function with FAT10. To date, it is known that a single E2 -USE1 is responsible for FAT10ylation. Here, the authors show a set of E2s that can be charged with FAT10 in vitro, using recombinant proteins, as well as in cells. The authors nicely show that FAT10 is getting charged on the active site of the E2, and that this does not happen with catalytically inactive mutants. Also, they show that these E2s require the last two amino acids of FAT10 (AV mutation prevents charging). Regarding cellular FAT10ylation, the authors show that treatment with TNF in 293 with USE1 KO gives the same FAT10ylation pattern as for cells expressing USE1. They then showed that some E2s could have the same effect as TNF. Specifically, the FAT10ylation pattern of cells lacking USE1 is similar whether treated by TNF or by overexpression E2s (for example, UBE2D3). Overall, the authors show for the first time that FAT 10 can function with other E2s. This observation by itself is very interesting. However, whether it is physiologically relevant is not clear. Showing that some E2s can be charged with FAT10 in cells lacking USE1 does not indicate whether it happens when USE1 is expressed. Specifically, showing that these E2 can also be charged with FAT10 in cells expressing USE1 would be more physiologically relevant. Also, using proteomics, can the authors show changes in FAT10ylation pattern between cells lacking and having a specific E2, that they claim functions with FAT10? Defining substrates whose FAT10ylation depends on another E2, that is not USE1, will highly support the author's conclusions.

The other point that is missing is the mechanism by which TNF upscales FAT10ylation. Since the authors did not find a significant increase in the expression of the E2s, the mechanism of TNF is probably not by increasing E2 expression. Therefore, the data show that overexpression of E2 can mimic the effect of TNF treatment, possibly not physiologically relevant. Instead of overexpression, can the authors generate double KO cells of USE1 and another E2 that will prevent TNF-inducing FAT10ylation? In other words, since USE1 is not needed for FAT10ylation upon TNF induction, can the authors find another E2 whose KO on the background of USE1 KO will prevent TNF from inducing FAT10ylation? This will significantly strengthen the manuscript and will be the basis for a detailed mechanistic study.

Reviewer #2 (Comments to the Authors (Required)):

The manuscript by Schnell et al focuses on deciphering the core elements required for transfer of FAT10 onto substrate proteins. FAT10 is a ubiquitin like protein and attachment to substrate proteins relies on E1, E2 and E3 enzymes. Here, the authors extend previous work to determine that an additional E2 enzyme can mediate the attachment of FAT10 to proteins and that this is particularly important in TNF stimulated cells. This conclusion is most strongly supported by Figure 1b which analyses attachment of FAT10 to proteins in a native setting i.e. no overexpression.

Following this discovery the authors seek to identify other E2s involved in FAT10 attachment, analysing transcriptomic data and then a collection of purified E2s. The manuscript contains considerable data aimed at identifying E2s and many of the individual experiments are carefully done. However, while they have identified additional E2s that can be charged with FAT10, the biological relevance of these E2s is uncertain. In part, I am reminded of the work of Kurz and colleagues Hjerpe R, Thomas Y, Kurz T (2012) NEDD8 Overexpression Results in Neddylation of Ubiquitin Substrates by the Ubiquitin Pathway. *J Mol Biol* 421: 27-29. This means that while charging of E2s with a range of ubiquitin like molecules (UBLs) is possible at high concentrations, it is difficult to determine which ones are biologically relevant.

Main points:

1) The analysis of putative additional FAT10 E2s in Figure 3 is difficult to interpret. While the authors show that the active site cysteines from both Ube2D3 and Ube20 can be charged with FAT10, the biological relevance is uncertain as the knockout experiments in panel e do not indicate a significant role. The authors then go on to analyse the ability of a number of other E2s to be charged with FAT10. They then show that when overexpressed these E2s can rescue substrate modification in USE1 KO cells. These experiments demonstrate the capacity of multiple E2s to be loaded with FAT10 but it is difficult to discern biological relevance. This is a significant issue. There might be a number of ways to address this but if the experiment reported in figure 1b was repeated in the different double knockout lines, with a readout of decreased FAT10 attachment, this would indicate that an

E2 was important.

Minor points:

- 1) Page 6 and other places they refer to 'FAT10 expression' when I think they mean substrate modification. This should be reviewed.
- 2) The order of panels in figure 3 is confusing. Panel e should be moved up as it I referred to after panel b.
- 3) One paragraph includes all material on pages 8-11. Better use of paragraphs should be considered throughout.
- 4) The manuscript needs a good edit throughout.

Point-by-point reply for manuscript #LSA-2023-01985-T

We are grateful to all reviewers for their constructive criticism and are convinced that our manuscript has benefited greatly from this revision. In the point-by-point reply below, we have answered all points, raised by our reviewers and hope that the new experiments and text changes will now satisfy their requests. We have exchanged the former title of the manuscript for the new title “Tumor necrosis factor mediates USE1-independent FAT10ylation under inflammatory conditions” because we think that this title describes better the revised version of the manuscript. Moreover, we have included a new co-author who helped with the revision and provided important new data. For an easier orientation, we have highlighted all main text changes in the Results and Discussion section in yellow.

Reviewer #1

In this manuscript, Schnell et al. identify E2 enzymes that function with FAT10. To date, it is known that a single E2 -USE1 is responsible for FAT10ylation. Here, the authors show a set of E2s that can be charged with FAT10 in vitro, using recombinant proteins, as well as in cells. The authors nicely show that FAT10 is getting charged on the active site of the E2, and that this does not happen with catalytically inactive mutants. Also, they show that these E2s require the last two amino acids of FAT10 (AV mutation prevents charging). Regarding cellular FAT10ylation, the authors show that treatment with TNF in 293 with USE1 KO gives the same FAT10ylation pattern as for cells expressing USE1. They then showed that some E2s could have the same effect as TNF. Specifically, the FAT10ylation pattern of cells lacking USE1 is similar whether treated by TNF or by overexpression E2s (for example, UBE2D3). Overall, the authors show for the first time that FAT 10 can function with other E2s. This observation by itself is very interesting. However, whether it is physiologically relevant is not clear. Showing that some E2s can be charged with FAT10 in cells lacking USE1 does not indicate whether it happens when USE1 is expressed. Specifically, showing that these E2 can also be charged with FAT10 in cells expressing USE1 would be more physiologically relevant.

We thank our reviewer for this comment and we completely agree with this suggestion. We have now repeated the loading experiments with UBE2A, -B, -C, D1, D3, G2, -O in HEK293 wild type cells, expressing decent amounts of endogenous USE1. The new data are shown in new Fig S3. As already observed in USE1-ko cells, FAT10 is loaded in a FAT10 diglycine- and E2 active site cysteine-dependent manner onto the respective E2 enzyme, also in the presence of USE1 in HEK293 wild type cells.

Also, using proteomics, can the authors show changes in FAT10ylation pattern between cells lacking and having a specific E2, that they claim functions with FAT10? Defining substrates whose FAT10ylation depends on another E2, that is not USE1, will highly support the author's conclusions.

We completely agree with this idea. The identification of FAT10 conjugation substrates which are dependent on one of the newly identified E2 enzymes is for sure very important. However, performing mass spectrometry (MS) analysis for seven E2 conjugating enzymes plus the subsequent verification of putative hits is a project that takes much more time and is not possible to perform within the three months of revision. Moreover, we have published several years ago a MS analysis of FAT10 conjugates (Aichele et al., JCS 2012) and have identified several hundred putative substrates, while most of them still await to be confirmed as FAT10 conjugation substrates. Thus, we think that it is beyond the scope of this manuscript to include such analyses. However, we indeed plan to identify E2-specific substrates in future experiments by using two different methods. First, we would like to perform a MS analysis using our newly generated USE1/E2 double-ko cell lines as well as the USE1-ko/siRNA knockdown cell lines in order to identify E2-specific FAT10 conjugation substrates. Moreover, we plan to establish the FAT10-ID method, which is related to the published SUMO-ID method (Baroso-Gomila et al., Nat Commun 2021 and Baroso-Gomila et al., Methods Mol Bio, 2023). All together, these analyses should result in the identification of many putative E2-specific FAT10 conjugation substrates which all then need to be confirmed in later experiments.

Apart from this, in new Fig 7 we present data showing that FAT10ylation of the E3 ligase ZNF598 is independent of USE1. We think that this finding strongly supports our data of additional E2 conjugating enzymes for FAT10ylation besides USE1. Moreover, we present α -synuclein as a new FAT10ylation substrate and show that the α -synuclein-FAT10 conjugate is dependent on USE1 when overexpressed in HEK293 cells. However, although it is absent in USE1-ko cells, it is strongly restored upon treatment of USE1-ko cells with TNF, and it is likewise enhanced in HEK293 wild type cells. Similarly, we observed FAT10ylation of the two E2 conjugating enzymes UBE2Q2 and UBE2QL1 (which turned out to be FAT10 conjugation substrates rather than E2 enzymes for FAT10ylation) only upon stimulation of USE1-ko cells with TNF, but never in untreated cells (Fig S2). We believe that these are very important data which strongly support our finding of a TNF-dependent FAT10ylation either in the presence or in the absence of USE1.

The other point that is missing is the mechanism by which TNF upscales FAT10ylation. Since the authors did not find a significant increase in the expression of the E2s, the mechanism of TNF is probably not by increasing E2 expression. Therefore, the data show that overexpression of E2 can mimic the effect of TNF treatment, possibly not physiologically relevant. Instead of overexpression, can the authors generate double KO cells of USE1 and another E2 that will prevent TNF-inducing FAT10ylation? In other words, since USE1 is not needed for FAT10ylation upon TNF induction, can the authors find another E2 whose KO on the background of USE1 KO will prevent TNF from inducing FAT10ylation? This will significantly strengthen the manuscript and will be the basis for a detailed mechanistic study.

We completely agree with this idea and have mentioned this already in the discussion. We have now generated CRISPR/Cas9-based double knockout cell lines for USE1/UBE2C, USE1/UBE2D3, USE1/UBE2G2, and USE1/UBE2O. In case of UBE2A, UBE2B and UBE2D1 we tried several times to generate knockout cells and used different gRNAs. However, we could not identify knockouts. We were facing the same problem before, when generating UBE2D3-ko cells. In this case we were successful because we finally found a UBE2D3-reactive antibody, which did not recognize UBE2D1, D2, or D4. We believe that we are facing the same problem in case of UBE2D1, UBE2A and UBE2B. For example in case of UBE2A it is even described in the respective antibody data sheet that the UBE2A antibodies recognize UBE2B as well, making it impossible to identify knockout cells. To solve this problem, we used specific siRNA directed against UBE2A, -B and -D1 and confirmed by real-time PCR that the mRNAs were downregulated by more than 90% (new Fig 7). However, as shown in new Fig. 7 and new supplementary Fig S5, the knockout or knockdown of all seven E2 enzymes together with USE1 had no impact on TNF-mediated FAT10 conjugation, pointing to the existence of at least one additional E2 enzyme that becomes active upon TNF treatment.

Concerning the mechanism of the TNF-mediated FAT10ylation, we do not want to rule out the possibility that another player such as for example an E3 ligase might be activated by TNF, rendering it capable to use one of the available E2 conjugating enzymes for FAT10ylation. We point to this possibility now also in the discussion where we now write:

“However, we also do not want to exclude the possibility that the TNF-mediated FAT10ylation might eventually be mediated not by a TNF-activated E2 enzyme, but maybe by another factor. For example, this be an E3 ligase which might be activated by TNF, rendering it capable to interact with a FAT10-specific E2 enzyme, other than USE1.”

Reviewer #2

The manuscript by Schnell et al focuses on deciphering the core elements required for transfer of FAT10 onto substrate proteins. FAT10 is a ubiquitin like protein and attachment to substrate proteins relies on E1, E2 and E3 enzymes. Here, the authors extend previous work to determine that an additional E2 enzyme can mediate the attachment of FAT10 to proteins and that this is particularly important in TNF stimulated cells. This conclusion is most strongly supported by Figure 1b which analyses attachment of FAT10 to proteins in a native setting i.e. no overexpression.

Following this discovery the authors seek to identify other E2s involved in FAT10 attachment, analysing transcriptomic data and then a collection of purified E2s. The manuscript contains

considerable data aimed at identifying E2s and many of the individual experiments are carefully done. However, while they have identified additional E2s that can be charged with FAT10, the biological relevance of these E2s is uncertain. In part, I am reminded of the work of Kurz and colleagues Hjerpe R, Thomas Y, Kurz T (2012) NEDD8 Overexpression Results in Neddylation of Ubiquitin Substrates by the Ubiquitin Pathway. *J Mol Biol* 421: 27-29. This means that while charging of E2s with a range of ubiquitin like molecules (UBLs) is possible at high concentrations, it is difficult to determine which ones are biologically relevant.

Main points:

1) The analysis of putative additional FAT10 E2s in Figure 3 is difficult to interpret. While the authors show that the active site cysteines from both Ube2D3 and Ube2O can be charged with FAT10, the biological relevance is uncertain as the knockout experiments in panel e do not indicate a significant role. The authors then go on to analyse the ability of a number of other E2s to be charged with FAT10. They then show that when overexpressed these E2s can rescue substrate modification in USE1 KO cells. These experiments demonstrate the capacity of multiple E2s to be loaded with FAT10 but it is difficult to discern biological relevance. This is a significant issue. There might be a number of ways to address this but if the experiment reported in figure 1b was repeated in the different double knockout lines, with a readout of decreased FAT10 attachment, this would indicate that an E2 was important.

We would like to thank our reviewer for this comment. We agree that overexpression might of course be a problem, however what argues against this idea is the finding that one E2 conjugating enzyme, namely UBE2C, was not able to reconstitute FAT10 conjugation in USE1-ko cells (Fig 6C and D), although it was clearly loaded with FAT10 onto its active-site cysteine (Fig 4C and D). However, it is indeed easier to discuss the biological relevance of the identified E2 enzymes for FAT10ylation when knowing specific target proteins, which are FAT10ylated by one of these E2 enzymes. Therefore, we plan to identify FAT10 conjugation substrates in future experiments by making use of two different approaches. First, we plan to perform a MS analysis using our newly generated USE1/E2 double-ko cell lines as well as the USE1-ko/siRNA knockdown cell lines in order to identify E2-specific FAT10 conjugation substrates. Moreover, we plan to establish the FAT10-ID method, which is related to the published SUMO-ID method (Baroso-Gomila et al., *Nat Commun* 2021 and Baroso-Gomila et al., *Methods Mol Bio*, 2023). Both analyses should result in the identification of FAT10 conjugation substrates, dependent on one of the identified E2 enzymes.

We have now generated CRISPR/Cas9-based double knockout cell lines for USE1/UBE2C, USE1/UBE2D3, USE1/UBE2G2, and USE1/UBE2O. In case of UBE2A, UBE2B and UBE2D1 we tried several times to generate knockout cells and used even different gRNAs but were not able to identify knockout cells. Therefore, we used specific siRNA to knockdown UBE2A, UBE2B, and UBE2D1 in USE1-ko cells. The USE1/E2 double knockout and USE1-ko/E2 knockdown cell lines were then used to investigate, if TNF-mediated FAT10 conjugation might be negatively influenced by the absence of the respective E2 enzyme, both, under overexpressing conditions with FLAG-FAT10 and under completely endogenous conditions upon induction of FAT10 expression with TNF and IFN γ . As shown in new Fig. 7 and new supplementary Fig S5, the knockout or knockdown of all seven E2 enzymes together with USE1 had no impact on TNF-mediated bulk FAT10 conjugation, pointing to the existence of at least one additional E2 enzyme that is activated upon TNF treatment. We additionally quantified the ECL signals of all three experiments with overexpressed FLAG-FAT10 (as shown in Fig 7A), what further confirms, that none of the E2 enzymes was involved in the TNF-mediated FAT10ylation in USE1-ko cells (Fig 1 of this point-by-point reply).

Fig 1. Quantification of ECL signals of FLAG-FAT10 conjugates of three independent experiments as shown in Fig 7A of the main manuscript. Values show the amount of FLAG-FAT10 conjugates normalized to the monomeric FLAG-FAT10 amount in the same cell lysate. The value of FLAG-FAT10 conjugates/monomeric FLAG-FAT10 in USE1-ko cells was set to unity and all other values were calculated accordingly (n=3).

Although the mechanism of how TNF is mediating FAT10 conjugation still needs to be explored in future experiments, we nonetheless were able to support our finding by the following new data (new Fig 7). We include now the identification of two new FAT10 conjugation substrates, which are FAT10ylated either independent of USE1, or dependent on TNF in USE1-ko cells. In new Fig 7, we show that the E3 ligase ZNF598 is FAT10ylated in a USE1-independent way, strongly supporting our finding of additional E2 enzymes besides USE1. Moreover, we present α -synuclein as new FAT10ylation substrate which depends on USE1 in the absence of TNF, but which is strongly FAT10ylated in the presence of TNF. We believe that this finding further supports our hypothesis of a TNF-mediated FAT10ylation, independent of USE1.

Minor points:

1) Page 6 and other places they refer to 'FAT10 expression' when I think they mean substrate modification. This should be reviewed.

We thank our reviewer for this comment but this might be a misunderstanding. We indeed mean protein expression and not FAT10 conjugation when we write "FAT10 expression". We have for example made the observation that FAT10 protein is always expressed to a lower amount in UBA6-ko cells, as compared to the FAT10 protein amount expressed in wild type cells. However, since anyway FAT10 conjugation is completely lacking in UBA6-ko cells, we have removed the following sentence from the manuscript (page 6): "Of note, endogenous FAT10 expression in UBA6-ko cells appears always less as compared to wild type cells."

2) The order of panels in figure 3 is confusing. Panel e should be moved up as it I referred to after panel b.

We agree that this is confusing and have changed the order in new Fig 3.

3) One paragraph includes all material on pages 8-11. Better use of paragraphs should be considered throughout.

We would like to thank our reviewer for this suggestion. We have now divided the large paragraph into three paragraphs.

4) The manuscript needs a good edit throughout.

We have made now all required changes and have included several new experiments. We hope that the revised version of the manuscript will now satisfy our reviewer.

August 10, 2023

RE: Life Science Alliance Manuscript #LSA-2023-01985-TR

Dr. Annette Aichem
Biotechnology Institute Thurgau
Unterseestrasse 47
Kreuzlingen 8280
Switzerland

Dear Dr. Aichem,

Thank you for submitting your revised manuscript entitled "Tumor necrosis factor mediates USE1-independent FAT10ylation under inflammatory conditions". We would be happy to publish your paper in Life Science Alliance pending final revisions necessary to meet our formatting guidelines.

- please add your main, supplementary figure, and table legends to the main manuscript text after the references section
- we encourage you to revise the figure legend for Figure 4 such that the figure panels are introduced in alphabetical order
- please add callouts for Figures S1A-B and SA-B to your main manuscript text;
- LSA allows supplementary tables, but not EV Tables; please update your callouts for the Supplementary Tables in the manuscript (Table S1, S2, etc.)

A. FINAL FILES:

B. MANUSCRIPT ORGANIZATION AND FORMATTING:

Sincerely,

August 11, 2023

RE: Life Science Alliance Manuscript #LSA-2023-01985-TRR

Dr. Annette Aichem
Biotechnology Institute Thurgau
Unterseestrasse 47
Kreuzlingen 8280
Switzerland

Dear Dr. Aichem,

Thank you for submitting your Research Article entitled "Tumor necrosis factor mediates USE1-independent FAT10ylation under inflammatory conditions". It is a pleasure to let you know that your manuscript is now accepted for publication in Life Science Alliance. Congratulations on this interesting work.

DISTRIBUTION OF MATERIALS:

Again, congratulations on a very nice paper. I hope you found the review process to be constructive and are pleased with how the manuscript was handled editorially. We look forward to future exciting submissions from your lab.

Sincerely,
